# Investigating the Impacts of Saharan Dust on Tropical Deep Convection Using Spectral Bin Microphysics, Part 1: Ice Formation and Cloud Properties

Matthew Gibbons<sup>1</sup>, Qilong Min<sup>1</sup>, Jiwen Fan<sup>2</sup>

<sup>1</sup> Atmospheric Science Research Center, State University of New York, Albany NY 12203, USA
 <sup>2</sup> Earth Systems Analysis & Modeling, Pacific Northwest National Laboratory

Correspondence to: Qilong Min (qmin@albany.edu)

Abstract. To better understand the impacts of dust aerosols on deep convective cloud (DCC) systems revealed by previous observational studies, a case study in the tropical eastern Atlantic was investigated using the Weather Research and

- Forecasting (WRF) model coupled with a Spectral Bin Microphysics (SBM) model as a two-part study. A detailed set of ice nucleation parameterizations linking ice formation with aerosol particles have been implemented in the SBM for this study. It is found that, dust, transported from the Sahara desert and acting as ice nuclei (IN), increases heterogeneous formation of ice particles at temperatures above -38°C by approximately a factor of four per IN magnitude increase from 0.12 cm<sup>-3</sup>. Homogeneous ice formation is reduced below -38°C by up to 79%, due to greater conversion of liquid drops to ice at warmer
- temperatures. The ice particle size distribution (PSD) is shifted towards smaller sizes at heterogeneous temperatures and median sizes at colder temperatures due to increased vapor competition and crystal aggregation. Graupel sizes are reduced due to increased riming of more numerous, but smaller, ice particles. Liquid mass is reduced by up to 85% at midlevels due to increased riming, drop freezing and Bergeron process evaporation. Despite the enhanced vertical motion in the dust cases (up to 30%), average cloud top height was found to be lowered by up to 3.29km in comparison with the background aerosol
- (Clean) case, which is consistent with observations. This is due to increased sedimentation rates resulting from earlier formation of precipitation sized particles.

#### **1** Introduction

Deep convective clouds (DCC) are important sources of precipitation and play a strong role in both regional and global circulation, with tropical convection being particularly significant (Arakawa, 2004). The strong updrafts within convective

- clouds can transport small cloud particles to the level of neutral buoyancy where they spread out to form the anvil associated with DCC (Folkins, 2002; Mullendore, 2005). The convective intensity controls the depth, area, and lifetime of the resulting anvil clouds (Futyan and Del Genio, 2007). The greater area coverage and lifetime persistence of the anvil cloud makes this type of cloud important to global energy balance and radiative transfer, making the study of deep convective clouds important for current and future climate research (Solomon et al., 2007), especially with regards to the highly uncertain
- effects of the aerosol indirect effect (AIE) mediated cloud radiative forcing (Rosenfeld et al., 2013).

Observational and modeling studies of DCC have shown different results relating to the effect of aerosol on convection and precipitation, indicating that aerosol may either enhance or suppress convection and precipitation depending on aerosol concentration and environmental conditions (Khain and Pokrovsky, 2004 Khain et al., 2004, 2005, 2008; van den Heever et al., 2006; Fan et al., 2007b; Min et al., 2009; Min and Li, 2010; Li and Min, 2010; Min et al., 2014; Altaratz et al., 2014). Clouds forming in elevated aerosol environments exhibit reduced cloud drop effective radii as a result of a greater number of smaller drops forming (Andreae et al., 2004; Koren et al., 2005), which in turn can result in less efficient collision-coalescence processes (Khain et al., 2005) delaying the formation of precipitation to higher levels in the clouds. Condensation and evaporation processes will be affected by the altered drop size distribution and number concentration, resulting in changes to the location and intensity of latent heat release within the cloud (Khain et al., 2005; Rosenfeld et al., 2008). Increased condensational latent heat release resulting from the greater droplet nucleation can result in stronger

- 2008). Increased condensational latent heat release resulting from the greater droplet nucleation can result in stronger convective updrafts leading to the formation of taller and wider clouds (Frederick, 2006; Zhang et al., 2007), while increased evaporation can result in stronger cold pool formation and enhanced secondary convection (Khain 2009, Lee et al., 2010). Aerosol related changes to cloud macrophysics are frequently attributed solely to convective invigoration by the increased cloud condensation nuclei (CCN). However, a study by Fan et al. (2013) involving simulations of DCC in three different
- regions, suggested that the observed taller and wider clouds are better explained by the changes of CCN to the microphysical properties. Thermodynamic invigoration did not unanimously occur in the study when polluted conditions were simulated, although increased cloud fraction and cloud top height were present. The study noted that the reduced hydrometeor sizes in the polluted case allowed greater cloud mass to be detrained from the convective core, and decreased particle fallout speed that slows down the cloud anvil dissipation.

5

Earlier studies tend to focus upon the action of soluble aerosols as cloud condensation nuclei (CCN), with changes to ice formation resulting from the affected liquid processes only (Rosenfeld, 2000). However, DCC can also be sensitive to the aerosols that act as ice nuclei (IN) (Tao et al., 2007). Ice formation in deep convective clouds may result from heterogeneous and homogeneous ice nucleation depending on the depth of the cloud and the chemical composition of the background aerosols. Heterogeneous ice nucleation can occur at temperatures between 0°C and -38°C via the mechanisms of condensation/deposition, immersion, and contact freezing (Tao et al., 2012; Altaratz et al., 2014; Hiron & Flossman, 2015) when ice nuclei (IN) are present. Homogeneous ice nucleation involves droplet and aerosol haze particle freezing at temperatures lower than -38°C. Deep convection frequently shoots liquid drops up to the upper troposphere where the temperature is colder than -38°C, leading to strong homogenous droplet freezing.

30

25

Dust aerosols have been observed at significant concentrations even in remote locations far from their expected source regions (Prospero, 1999) and are predominately composed of insoluble silicate particles (Lohmann, 2002) which have been established to act as effective IN (Pruppacher and Klett, 1997; Demott et al., 2003; Sassen et al., 2003). The Saharan Air Layer (SAL; Prospero and Carlson, 1970; Carlson and Prospero, 1972) is an elevated layer of dry air between 850-500 hPa,

often containing lofted dust particles. The SAL has been observed interacting with tropical cloud systems, such as tropical cyclones and mesoscale convective systems (MCS), and may impact their intensity and evolution (Karyampudi and Carlson, 1988; Dunion and Velden, 2004; Min et al., 2009). Ekman et al. (2007) studied the sensitivity of a continental storm to IN concentration and found that updrafts were enhanced due to added latent heat release from ice crystal depositional growth.

- The stronger updrafts enhanced homogeneous nucleation, increasing anvil cloud coverage and precipitation. Fan et al. (2010) compared the effects of CCN and IN on convection and precipitation and noted that the CCN effect is more evident in changing cloud anvil size, lifetime, and microphysical properties. IN was shown to have a small effect on convective strength, but the microphysical effects could still be significant. Note Fan et al. (2010) did not have a prognostic IN treatment as what we have done for this study.

A trans-Atlantic dust outbreak of Saharan origin occurring 1-10 March 2004 (Morris et al., 2006) was subjected to a rigorous multi-sensor and multi-platform observational analysis (Min et al., 2009; Li et al., 2010; Min and Li, 2010; Li and Min, 2010; Min et al., 2014). The interaction of this dust outbreak with developing DCC systems resulted in strong effects on cloud microphysical processes. The effects of dust resulted in changes to the vertical precipitation structures, in which the

- size spectrum was shifted from heavy to light precipitation or suppressed entirely (Min et al., 2009; Li and Min, 2010). These microphysical changes can alter the vertical heating structure in both the convective and stratiform regions, possibly creating strong cloud thermodynamical feedbacks (Rosenfeld et al., 2008). When cloud dynamics and thermodynamics are strictly constrained, the effects of dust on warm cloud processes are strongly dependent on cloud top height and precipitation regime, indicating that vertical transport is important to aerosol-cloud interactions (Li et al., 2010). It was also found that
- substantial changes to cloud top distributions resulted from a change in the partition of homogeneous and heterogeneous ice processes under dusty conditions, with such macrophysical changes in the cloud systems resulting in substantial thermal infrared radiation cooling of up to 16 Wm<sup>-2</sup> (Min and Li, 2010). In this study, we will focus on how dust in SAL impacts ice cloud microphysical processes in DCC and the resulting change to cloud macrophysical and precipitation properties. Convective invigoration has been observed under dusty conditions, which can enhance water vapor supply in precipitating
- DCC and lift ice particles to higher altitudes. These microphysical effects in concert with large scale dynamics and thermodynamics influence macrophysics of DCC and associated precipitation. (Min et al., 2014).

The quantification of aerosol-cloud interactions in numerical modeling studies remains a challenge. Aerosols influence the character of individual clouds and storms, but evidence of a systematic effect on storm or precipitation intensity is still

limited and ambiguous. The representation of processes relevant to aerosol-cloud interactions is still considered weak, due to the some of the fundamental details of cloud microphysical processes being poorly understood, particularly with regards to ice and mixed-phase clouds (Boucher et al., 2013). This low confidence is a result of the complex coupling between the processes controlling cloud and precipitation properties, which cover a wide range of spatial and temporal scales (Tao et al., 2012). To gain better insight into mixed-phase cloud and precipitation processes, numerical models must include a detailed

handling of ice microphysical processes that connect ice formation to the presence of ice nuclei, such as atmospheric dust. The study of Halder at al. (2015) tested the sensitivity of severe thunderstorms to different bulk microphysics and found the best agreement with observations occurred when aerosol-based heterogeneous ice formation was added to the most detailed bulk microphysics scheme.

By selecting bin microphysics rather than bulk microphysics, it is possible to explicitly solve the relevant microphysical equations without making a priori assumptions about the particle size distribution (PSD) of the resulting hydrometeors or modifying the scheme to simulate different atmospheric phenomena from tropical to polar environments (Khain et al., 2015). Unlike a bulk microphysics scheme, in which the PSD of every predicted hydrometeor type has a predefined shape, bin microphysics schemes allow PSDs to evolve naturally in conjunction with in-cloud and large scale processes. To take advantage of this greater accuracy, we have selected the Hebrew University Cloud Model (HUCM) Spectral Bin Microphysical (SBM; Khain et al., 2004; Khain et al., 2009; Fan et al., 2012a) for use in our study.

As dust aerosols are widely known to affect heterogeneous ice formation, accurate simulations of ice formation processes in DCC require ice nucleation to be directly linked with aerosol concentration. In this study, we have added a set of ice nucleation parameterizations to the WRF-SBM to examine how dust acting as IN impacts cloud properties and precipitation. Heterogeneous formation of ice resulting from immersion, contact, and deposition-condensation freezing schemes have been updated to account for the presence of dust in our case study. These updates follow the implementation first suggested in Gong et al. (2010), with additional schemes and updates incorporated from Fan et al. (2014). Specific updates made to the

- model have been outlined in section 2. The broad coverage provided by these multiple ice formation schemes in conjunction with the detailed handling of cloud liquid and ice microphysics offered by the WRF-SBM gives our study a unique insight into the topic of dust aerosol-DCC interactions. To assess the agreement of the model with observations, we have simulated a MCS occurring on 08 March 2004 in the tropical eastern Atlantic under the influence of a Saharan dust outbreak (Morris et al. 2006; Min et al. 2009). The results of our study have been presented as two parts. The first part, introduced in this paper,
- outlines the changes made by our group to the WRF-SBM coupled model and includes an overview of the results pertaining to the effects of dust on ice formation and resulting cloud properties. The second part will focus on the resulting changes to the precipitation fields under dusty conditions.

#### **2 Model Description**

Numerical simulations were undertaken using the WRF version 3.1.1 developed by the National Center for Atmospheric 30 Research (NCAR) as described in Skamarock et al. (2008). WRF solves the fully compressible, non-hydrostatic Euler equations formulated on terrain following hydrostatic-pressure coordinates and the Arakawa C-grid. The model uses Runge-Kutta second- to sixth-order advection schemes in both horizontal and vertical directions. The fifth-order advection scheme

is used in this study. The monotonic technique is employed for advection of scalar and moist variables. The cloud microphysical scheme is described below.

#### 2.1 Spectral Bin Microphysics (SBM)

- The original SBM (Khain et al., 2004) solves a system of kinetic equations for the size distribution functions for 7 5 hydrometeor types: water droplets, ice crystals (plate, column, and dendrite), aggregates, graupel, and frozen drops/hail. An 8th size distribution function exists for CCN. Each size distribution is represented by 33 mass doubling bins, where the mass of a particle in each bin is twice the mass of a particle in the preceding bin. A fast version of the SBM (Fast-SBM) with four size distributions of water drops, low density ice (ice crystals and aggregates), high density ice (graupel and hail), was created in order to substantially reduce the computational costs (Khain et al. 2009, Fan et al 2012). Further details about the
- mechanics of the SBM can be found in Khain et al. (2004) and Fan et al. (2012) and will not be repeated here.

In order to examine IN impacts on clouds and precipitation, an additional prognostic variable for IN particle (dust in this case) concentration was added to represent the sink by ice nucleation and sources of dust (from lateral boundaries). In this study, due to the presence of a dust layer in the observed case, such a layer has been added. The dust in the layer can serve as IN, CCN, or some fractional combination of the two can be set by the user. This allows us to test the sensitivity of clouds

IN, CCN, or some fractional combination of the two can be set by the user. This allows us to test the sensitivity of clouds within our model to such a mixture of nuclei. In this study, we primarily consider the dust effects by acting as IN, but also have run a preliminary sensitivity study allowing some dust to act as CCN in addition to IN.

#### 2.2 Ice Formation Parameterizations

The original SBM (Khain et al. 2004) included both homogeneous and heterogeneous ice formation, although without 20 directly connecting ice formation to a prognostic IN variable. Liquid drop freezing for both homogeneous and immersion mechanisms was provided by Bigg (1953). Ice formation resulting from condensation and deposition freezing was provided by Meyers et al. (1992). Contact freezing is not included in the original SBM. In order to perform a study of aerosol impacts on heterogeneous ice formation, it is necessary to directly link ice nucleation rates to aerosol properties. The study of Gong et al. (2010), and more recently Fan et al. (2014), updated the available homogeneous freezing mechanisms and additionally

- implemented separate parameterizations into the SBM for depositional, contact and immersion freezing, with ice formation in each of these schemes directly linked to the prognostic IN variable. This implementation also tracks the individual contribution of each heterogeneous and homogeneous freezing mechanism to the total ice formation, allowing for these contributions to be compared under different dust loading conditions. To take advantage of this unique and comprehensive set of ice formation parameterizations, we have adopted and expanded upon the Gong et al. (2010) homogeneous and homogeneous ice formation mechanisms into our surrent study, with available schemes outlined below.
- heterogeneous ice formation mechanisms into our current study, with available schemes outlined below.

5

15

#### 2.2.1 Homogeneous Ice Nucleation and Freezing Schemes

The new ice formation implementation allows the user to select between three available pure drop freezing schemes. We maintain the parameterization of Bigg (1953), included in the original SBM code, to serve as a benchmark for homogeneous pure drop freezing. We have also incorporated the parameterization of Heymsfield and Milosevich (1993), implemented in Gong et al. (2010), and a simple (temperature threshold only) based freezing scheme ( $H_{thr}$ ) to provide alternate methods of homogeneous pure drop freezing. The  $H_{thr}$  threshold based scheme automatically freezes all liquid drops that reach the -38°C homogeneous freezing threshold temperature, irrespective of drop size, under the assumption that liquid drops are rare under these conditions. These alternate methods can be used to replace the original Bigg (1953) parameterization in the simulation, in order to study the differences between the original stochastic (Bigg) and the newer temperature and/or drop-volume

10 (Heymsfield and Milosevich;  $H_{thr}$ ) based freezing rates. The current results use the  $H_{thr}$  threshold based scheme to provide pure drop freezing.

We also include the aerosol freezing parameterization of Liu and Penner (2005), following the implementation used by Gong et al. (2010), allowing for cirrus cloud formation to be studied. This parameterization calculates the homogeneous ice nucleation of aerosols coated by sulfate, while also including depositional nucleation as originally described by Meyers et al. (1992). In addition to the pure sulfate aerosol number concentration, the scheme includes the effects of updraft velocity (w)

and temperature under a critical relative humidity threshold with respect to water when calculating new ice formation.

#### 2.2.2 Heterogenous Ice Nucleation and Freezing Schemes

Depositional/condensational freezing in the model were originally provided by the Meyers et al. (1992) parameterization. It 20 is a supersaturation dependent parameterization. Deposition nucleation should be important for deep convection, but there is 20 no such a scheme developed for deep convection with a connection with aerosols. As noted in Meyers et al. (1992) it is 20 difficult to distinguish the relative contributions of depositional and condensational freezing in a parcel, since both form 20 similarly sized small ice crystals, despite the different mechanisms of vapor to ice in the former and condensation followed 20 immediately by freezing in the latter case. However, studies suggest that small ice formed in the temperature range covered

by this scheme can have a large impact on subsequent ice formation at higher altitudes (Ackerman et al. 2015; Hiron and Flossman, 2015; Lawson et al., 2015). To link depositional/condensational freezing with aerosols, we follow the implementation of van den Heever et al. (2006), updated from the Meyers et al. (1992) version. The number of ice crystals generated by depositional-condensational nucleation ( $N_{hen}$ ) is proportional to the IN number concentration ( $N_{IN}$ ;  $\Gamma^1$ ) by Eq. (1):

30

$$N_{hen} = N_{IN}F_M,\tag{1}$$

where  $F_M$  (unitless) is the function of the depositional-condensational nucleation by Meyers et al. (1992) that represents the fraction of available IN (Nid;  $1^{-1}$ ) as calculated in Eq. (2):

$$N_{id} = exp\{-6.39 + 0.1296[100(S_i - 1)]\},\tag{2}$$

with  $S_i$  being the saturation over ice. The value of  $F_M$  is equal to 1 for conditions at ice supersaturation of 40%, at which point all IN are activated, and is equal to 0 when supersaturation over ice is negative. The initial size of an ice crystal formed by this scheme is assumed to be 2.5 µm in radius and is assigned to the smallest ice size bin.

As stated above, the immersion freezing in the original scheme uses the parameterization of Bigg (1953), which is temperature-dependent only. To provide an aerosol-based immersion freezing scheme, we have incorporated the parameterization of DeMott et al. (2015), which was implemented by Fan et al. (2014) (cited as DeMott et al. (2013) in Fan et al. (2014) due to DeMott et al. (2015) not yet being published). The DeMott et al. (2015) immersion freezing rate is parameterized as in Eq. (3):

$$N_{im} = (CF)(N_{IN})(\alpha[273.16 - T_k] + \beta)exp(\gamma[273.16 - T_k] + \delta),$$
(3)

CF is an instrumental correction factor with a value of 3. Coefficients α, β, γ, and δ are 5.95E-5, 1.25, 0.46, and -11.6, respectively, representing mineral dust particles (Demott et al., 2015) T<sub>k</sub> is the cloud temperature in degrees Kelvin, N<sub>IN</sub> is
the number concentration of total aerosol particles with diameter larger than 0.5 µm, and N<sub>im</sub> is the maximum number of immersion ice possible in the given temperature range. Liquid drops are consumed over the size spectrum starting with the largest sizes down to the smallest until the minimum of N<sub>im</sub> or drop number is reached. According to Yin et al. (2005), drops with a radius smaller than 79.37 µm will be frozen to pristine ice crystals, otherwise graupel is formed.

- We have also adopted the contact freezing parameterization of Muhlbauer and Lohmann (2009), which is based on Cotton (1986) and Young (1977). In this parameterization, contact freezing is a result of the collision of supercooled liquid water drops and IN due to Brownian motion. The contact freezing rate is therefore proportional to the drops' radius and number concentration. It is also proportional to the IN number concentration and Brownian diffusivity in air. Unlike Muhlbauer and Lohmann (2009) who calculated the freezing rate for the sum of all drops, we perform the calculation in this study for each
- spectral bin of drops. Then, the contact freezing rate ( $N_{CNT}$ ;  $l^{-1}s^{-1}$ ) for each individual size bin is represented by Eq. (4):

$$N_{CNT} = 4\pi r_c N_c D_k N_{IN},\tag{4}$$

where  $r_c$  (m) and  $N_c$  (m-3) is radius and number concentration of drops respectively.  $D_k$  is the dust aerosol Brownian diffusivity (m2s-1), and is parameterized by Eq. (5):

$$D_k = \frac{k_B T C}{6\pi \eta r},\tag{5}$$

 $D_k$  is a function of the Boltzmann constant  $K_B=1.28 \times 10^{-23} \text{ m}^2 \text{ kg s}^{-2} \text{ K}^{-1}$ , T is the air temperature, r is the dry dust aerosol median radius,  $\eta$  is the viscosity of air and C is the Cunningham slip correction factor. The viscosity of air depends on temperature, as calculated by Eq. (6):

$$\eta = 10^{-5} [1.718 + 4.9x 10^{-3} (T - 273.15) - 1.2x 10^{-23} (T - 273.15)^2],$$
 (6)

The Cunningham slip correction factor is calculated by Eq. (7):

$$C = 1 + 1.26 \frac{\lambda}{r} \frac{1013.25}{p} \frac{T}{273.15},\tag{7}$$

with the molecular mean free path length of air  $\lambda$ =0.066 µm, and p is the pressure. To simplify the calculation, the contact freezing number is the available dust number concentration N<sub>IN</sub>, with freezing efficiency of 1. Upon freezing, drops with a radius smaller than 79.37 µm will be frozen to pristine ice crystals, larger drops will be frozen as graupel.

It should be noted that currently there is no ice nucleation parameterizations specifically developed for DCC, and the understanding of ice nucleation for DCC is still very limited. The most recent ice nucleation parameterization of DeMott et al. (2015) in conjunction with the previously stated parameterizations have been used in this study to gain the understanding of dust aerosol impact on DCC by serving as IN and the relative contribution of each mechanism to the total ice formation.

## **3 Experiment Design**

- For our study, we have conducted experiments simulating the 08 March 2004 MCS described in Min et al. (2009), using realistic initial and boundary conditions. Four one-way nested domains were used, with horizontal grid resolutions of 81km, 27km, 9km, and 3km respectively and 41 vertical levels in each domain. The numbers of horizontal grid points in each domain are 81x81, 81x81, 81x81, and 150x150, respectively. Initial and boundary conditions for the first domain are provided by the 1° x 1° 6-hourly National Centers for Environmental Prediction (NCEP) global final analysis dataset, with
- initial conditions for the other three domains being interpolated from the first domain. Due to the SBM not being designed to run at coarse resolutions, the SBM provides microphysics for only the 3km resolution domain with bulk microphysics being

selected for domains 1-3. The specific WRF parameterizations selected for the experiments are detailed in Table 1. Figure 1 provides the locations of these four domains, displaying the Atmospheric Infrared Sounder (AIRS) retrieval (Fig. 1a) and the model derived precipitable water averaged over the duration of the simulation (Fig. 1b). The large scale patterns are well reproduced by the model, assuring that meteorological conditions in the region of interest are being represented correctly.

5

10

30

The initial number concentrations of CCN are kept identical between the different cases. Typical marine aerosol number concentrations tend to be low, on the order of 300-600 cm<sup>-3</sup> (O'Dowd et al., 1997; Yoon et al., 2007). Therefore, the CCN number is set to a uniform value of 300 cm<sup>-3</sup> below 2km with the CCN number being reduced exponentially from this value as height increases above 2km. The initial IN distribution is set to be vertically uniform at .01 cm<sup>-3</sup> for the Clean case. The dust cases add an increasing number of IN to the Clean case's background value in a layer located vertically between 1km and 3km, as observed by Min et al (2009). The dust cases are set with different IN numbers within the dust layer of 0.12 cm<sup>-3</sup> (case D.12), 1.2 cm<sup>-3</sup> (case D.12), and 12 cm<sup>-3</sup> (case D12), respectively. These values were selected based on aerosol measurements that were taken during the trans-Atlantic Aerosol and Ocean Science Expeditions (AEROSE) experiment

(Morris et al., 2006) for dates coinciding with the observational study of the March 2004 dust outbreak detailed in Min et al.

- (2009). The dust loading was assumed to be the difference in the aerosol number of the dusty and pristine periods. Only aerosol particles with a radius greater than 0.5 microns were considered when taking this difference, due to the smaller aerosol sizes being more prevalent during the pristine period compared to the dusty period; a size range also consistent with the study of DeMott et al. (2015) for ice nucleating particles. This resulting dust number was multiplied by an activation fraction suggested by Niemand et al. (2012) for Saharan dust to arrive at the number used for case D.12. Other studies have
- suggested that dust related IN numbers greater than 1.0cm<sup>-3</sup> are possible (DeMott et al., 2003; Sassen et al., 2003), so two additional dust cases with IN numbers one (D1.2) and two (D12) orders of magnitude greater than the initial D.12 case were included in the study. To prevent the CCN and IN fields from being diluted due to the inflow of air from the lateral boundaries, the CCN and IN numbers of the outer five grid cells (i.e., the boundary points) on each side of domain 4 are set to the initial values throughout the integration period. The initial vertical profile of domain averaged relative humidity shows
- moist (>60% RH) air below 6km and drier air (

classified as precipitating column with vertical motion exceeding a 1m/s threshold and cloud thickness of 8 km or greater. Non-convective precipitating columns are classified as stratiform by the presence of ice-phase precipitation in the column. Precipitating columns with cloud top temperatures warmer than freezing are classified as rain producing warm clouds.

#### **4 Results**

#### 5 4.1 Cloud Geometry and Vertical Motion

The study of Min et al. (2009) found the evidence of dusty conditions increasing the frequency of deep clouds with large ice water path, hypothesizing that the action of dust as IN served to increase convective activity due to heterogeneous ice formation increasing latent heat release. To provide an initial comparison with these results, Fig. 2 depicts a vertical cross-section of a specific convective core and its associated stratiform/anvil cloud at a single model time step from the Clean and

- 10 D1.2 cases. The grey dashed line in Fig. 2a-2d depict the threshold value of cloudiness suggested by Fan et al. (2013) and shows the change to cloud geometry directly. The Clean case (Fig. 1a) shows the classic DCC cloud structure of convective core and associated stratiform region transitioning into the anvil. The D1.2 case also possesses a similar cloud structure, but with a far smaller anvil cloud, which can be a result of either dynamical or microphysical changes. The change in total water content (TWC) in Fig. 2a and Fig. 2b and total (ice and liquid) rain rate in Fig. 2c and Fig. 2d are consistent with the findings
- 15 of Min et al. (2009). The higher 0°C to -38°C TWC is accompanied by a correspondingly larger area of strong vertical motion as outlined by the black solid (>1m/s) and dashed blue (