# Peer review of "Investigating the Impacts of Saharan Dust on Tropical Deep Convection Using Spectral Bin Microphysics, Part 1: Ice Formation and Cloud Properties"

_Atmospheric Chemistry and Physics, 2016_

## Referee Comment (RC1) · Anonymous Referee #1 · 28 Jun 2016

General Comments: This paper investigates the impact of dust acting as IN on tropical convection – specifically its impacts on ice nucleation and particle size distributions - using numerical simulations. The impacts of dust on tropical convection are still not well understood and studies such as this one are needed. Some aspects of the analysis and discussion need clarification, particularly regarding Figure 7 and some of the physical reasoning for the differences seen in the simulations. These are detailed below.

Specific Comments: 1. Page 2, Line 10: It is confusing to state the increased condensation results from greater droplet nucleation since those are two separate processes.

[Figure]

More accurately, the higher droplet concentrations induce the greater condensation and heat release.

2. Page 5, Lines 12-17: Hasn't the additional IN prognostic variable been available in the SBM for a number of years now?

3. Page 6, Line 10: Why is the simplest homogeneous nucleation option being used? Presumably the other options represent homogeneous nucleation more accurately.

4. Page 8, Line 20: If the authors feel that our ability to represent ice nucleation in models is limited and poor, how do we know that the results of this study are meaningful and applicable to the real world?

5. Page 10, Lines 20-25: I found this part to be confusing. Are the authors simply trying to state that differences in simulated cloud properties are entirely due to dust impacts? I think it is a given that the environment and sea surface are initially the same and therefore do no contribute to differences in the clouds.

6. Page 11, Line 9: Max updraft speed or mean updraft speed?

7. Page 11, Lines 14-to end of page: The logic here is circular. First the authors state that the weaker updrafts limit ice nucleation (also, why is this the case?), but aren't the changes in nucleation ultimately the reason for the weaker updrafts?

8. Page 12, Line 12: What is meant by "three per unit magnitude increase"?

9. Page 13, Line 13: What is meant be "increases up to 30%, proportionally to IN number"?

10. Page 13, Line 25-26: Does this ratio appear in your simulations?

11. Page 14 and Figure 7: I don't understand what is shown in Figure 7. The authors state it is the difference in location before and after gravitational sedimentation. Are both of these fields output by the model? The titles on the figures say "number flux" but the units are 1/L which is not a flux. Lastly, the text suggests that the authors examine

this quantity in order to understand how particles are transported, but are the particles not also transported by the winds? And thus this figure does not really tell us where the regions of formation are?

12. The authors distinguish between heterogeneous and homogeneous ice throughout the paper. This is based just on the air temperature? But it is possible for heterogeneously nucleated ice to be transported higher in the atmosphere where homogeneous nucleation is dominant, yes? And likewise homogeneously nucleated ice can fall to lower levels. It seems to me that the two types of nucleated ice can't be easily distinguished and that the labels are perhaps misleading.

13. In Figures 5 and 6, I understand why showing values on a log scale is useful, but I don't understand why the authors add 10 – this just makes the values more difficult to interpret. Also, it is worth pointing out that in the difference of two log10 values is the log10 value of the ratio, i.e. $\log(x)-\log(y)=\log(x/y)$, in order to give more physical meaning to these plots.

14. The changes in relative importance of nucleation mechanisms is interesting. I would suggest moving this discussion to earlier in the results section. Changes in nucleation is the first step in the chain of events that lead to the changes in cloud properties, so it seems natural to include this discussion first rather than last in the results section.

15. There have been several studies examining the impacts of dust on tropical convection, particularly in hurricanes, yet in general only those studies by Min et al. are cited. Better citation of other relevant literature is needed.

16. The lower cloud top heights despite stronger updrafts is a bit confusing, though the authors do give some reasons. I'm wondering if the cloud tops are lower only in the stratiform regions, whereas the strongest updrafts are of course in the convective cores? Perhaps the cloud tops in the convective cores are more similar?

---

## Referee Comment (RC2) · Anonymous Referee #2 · 29 Aug 2016

General Comments:

The manuscript addresses several interesting hypotheses based on observations from a group of papers [Min et al. (2009), Min and Li (2010), and Li and Min (2010)] regarding the impacts of dust particles on the characteristics of tropical deep convection via nucleation pathways. Using a suite of simulations that vary the dust concentrations and the hydrometeor activation pathways for one specific case from these observational papers, the authors assess the impact of dust aerosol on ice and cloud nucleation, and subsequent microphysics. The topic of this paper makes it suitable for ACP, and the

use of spectral bin microphysics make it unique, however, there are numerous short-falls in the manuscript that currently make it unsuitable for publication in ACP, including (1) a lack of analysis and/or justification for a number of the statements made; (2) the absence of number relevant citations as well as placing the current research within the context of past work; (3) little to no demonstration regarding how well the case study output compares with the observations; (4) issues related to the convective-stratiform partitioning and the convective thresholds; (5) little to no substantiated physical reasoning in the role of the dynamics; and (6) poor figure quality. These are outlined in detail below.

Introduction and References

1) P1 L27: In addition to convective intensity, aerosols have also been found to influence the anvil characteristics as described in papers by Fan et al (2007) and a recent paper by Saleeby et al (2016).

2) P2 L4-19: The authors have cited a lot of their own work, which is fine, however, I would encourage them to cite other relevant work here too. For example, there are numerous other observational and modeling papers examining the impacts of Saharan dust on deep convective systems including MCSs and hurricanes that are relevant to this work that have not been cited including Dunion and Velden (2004), Evan et al (2006), Lau et al (2009), Zhang et al (2009), Braun (2010), Carrio and Cotton (2011), Cotton et al (2012), Storer and Van Den Heever (2013) and Storer et al (2014). Some of these references are also relevant to the statements made on page 3.

3) P2 L12: There are numerous observational and modeling papers that have shown that cold pools are warmer in more polluted environments (Altaratz et al. 2007, Berg et al. 2008, Storer et al 2010, Lim et al. 2011, May et al. 2011, Morrison 2012, Grant and Van den Heever, 2015). These papers argue that raindrops are larger under more polluted conditions due to greater rain accretion of the higher cloud water contents produced in more polluted conditions. Please be sure to include this argument too in

your description of cold pool impacts.

4) P2 L21: It should be notes that dust can serve as CCN and/or IN as shown by various papers by Twohy et al (2009 and others)

5) P2 L23: The paper by Van den Heever et al (2006) which the authors cite earlier, was one of the first to investigate the impacts of both CCN and IN on DCCs and should be cited here.

6) P2 L26: While these are all fine references cited here, these papers did not "discover" heterogeneous nucleation per se. The authors should go back to the original statements on these processes, or alternatively state what the papers referenced here added to the field of heterogeneous nucleation.

7) P3 L24-25: The past research of Koren et al (2005, 2009) and Wall et al (2014) also utilized observations to examine convective invigoration in response to dust across the globe.

Experiment Setup

1) P5 L12-17: "sources of dust (from lateral boundaries)" – are the lateral boundaries the only sources of dust in this model setup? Please clarify. Also, it is stated that "such a layer has been added". How was this later added? What are the characteristics of this layer? Finally, how is the fractional combination of dust serving as CCN and IN parameterized? And what was it set to in these simulations?

2) P6, L10: Why did the authors choose to use the Hthr homogenous nucleation method over the other two methods described? Furthermore, what is the point of describing the H&M method if it is simply an addition to the model that was not utilized in this study?

3) P6 L20-21 (and later P8 L20): The authors state "Deposition nucleation should be important for deep convection, but there is no such a scheme developed for deep convection with a connection with aerosols." They then go on to state in the same

paragraph "To link depositional / condensational freezing with aerosols, we follow the implementation of van den Heever et al (2006), update from the Meyers et al (1992) version." It would thus seem that such a scheme has been developed before. Furthermore, from what I recall, such a scheme does exist in the RAMS model used in the Van den Heever study. Do the authors mean that no such scheme exists in this version of the SBM? Please clarify this in the text.

4) P6, L24: Please specify temperature range.

5) P6-P7, Eq (1-2): Are Nid and NIN the same variable? Can you please clarify?

6) P7-8, Eq (4-7): Is the explanation of how NCNT is calculated necessary if the authors are just keeping it set at a constant value?

7) P8, Eq (7): State what "r" stands for in Eq 7.

8) P8, L27: What is the grid spacing for the 41 vertical levels in the simulations? Also, 3km and 41 vertical levels seems relatively coarse for resolving the updrafts and cloud processes of MCSs. Have the authors tested the sensitivity to grid resolution in any way? This could be tested with a less costly bulk scheme to ensure that enhancing grid resolution doesn't have a significant impact on the case study results.

9) P9, L3: While the large scale precipitable water patterns are relatively well produced, the magnitudes are quite different. This could have a significant impact on both the amount of liquid water and ice formed by the model. Please can the authors comment on this?

10) P16, L2: Adding 12 CCN cm-3 to the dust layer for case D1.2c seems like a somewhat arbitrary number, especially given that the numbers measured were more than double this ($\sim$30/cc) as stated in the paper. Why not add 30/cc? Also, surely a more correct approach here is not to add more CCN, but rather attribute some portion of the initial dust population to CCN and some to IN, and hence remove the number that are allowed to act as CCN from the number that were allowed to operate as IN in the first

set of simulations? In the current approach, the authors have simply added more dust particles to the environment. This does not appear to represent the problem correctly, in that when dust can act as CCN or IN, when they operate as CCN this prevents them from acting as IN (unless through immersion freezing) as they are typically rained out. In the current approach, the simulations are simply being made more polluted, and hence the effects of greater dust concentrations, as opposed to a different partitioning between CCN and IN, is being investigated. Please will the authors comment on this approach?

Case Study Evaluation

1) P9, L3: The authors motivate this study and discuss the results of this study based on a suite of observational studies by Min et al. (2009), Li and Min, (2010), and Min and Li (2010). However, there is not enough analysis presented in this study to demonstrate whether the simulations are properly reproducing the MCS event on 08 March 2004. The only comparison to observations is a precipitable water comparison, which is averaged over a 3-day period. While this does provide some information about the large-scale environment, it does not provide enough information on whether the convective elements and environment are similar to those observed and discussed in the observations studies listed above. Many studies have shown that aerosol effects on convection may depend on the convective system structure and environmental conditions, some of which are cited in this document. Therefore, the authors should include additional comparison of the simulations with observations in terms of the environment and convective system structure, especially since the authors are trying to explain an observed phenomenon represented by these simulations. Given that the observations from this case study have been published by some of the authors, it seems reasonable to compare some of the observations put forth in these prior studies to the same values in the simulations.

2) Pg line 11: The authors state that dust number concentrations were enhanced within the layer between 1 and 3 km. What did the authors do for the regions not included

within the SAL? What were these concentrations set to? The SAL would not realistically extend across the entire region captured by their grid 1. Also, the authors state that these concentrations are those for IN. Can these particles also serve as CCN? The authors refer to a relative combination of serving as CCN and/or IN earlier in the paper, however, that is not indicated here.

3) P9 line 12: There seems to be an error in one of these case study names as two of them are the same.

4) P9, L24-26: The SAL is typically a dry layer, with layers that tend to be moister below and above the SAL. Why is the dryness of SAL not represented here? Also, what data are used to provide the initial environmental data, as well as the boundary conditions?

Scientific Analysis, Reasoning, Clarity and Conclusions

1) P1 L15: "Ice particle size distributions" – which ice particles are the authors referring to? Smaller ice, larger ice, all of the ice species? As ice can refer collectively to small and large ice species, this should be clarified.

2) P1 L20: "which is consistent with observations" – which observations are these? Do the authors mean observations of the current case? Or to observations more generally. If the latter, other observations have not always found this to be the case.

3) P2 L13-16: There are various definitions of convective invigoration ranging from stronger updrafts, to higher cloud tops, to more precipitation production to impacts on latent heating, with some of these being related. I would recommend that the authors make clear what they mean by convective invigoration, thermodynamic invigoration etc.

4) P4 L9: Do note that some bulk schemes diagnose the shape parameter.

5) P10, L1-3: The authors partition precipitating columns into convective and stratiform regions, using somewhat arbitrary thresholds for vertical motions and cloud depth. Furthermore, there seems like there may be a few loopholes in their classification. For example, if a column has cirrus clouds in the upper troposphere, but precipitating, shallow

warm-phase clouds below it, it seems like these would be classified as stratiform. Is this correct? If not, and these are classified as warm clouds, then the upper level cirrus would be excluded from their analysis. An example of such an instance can be seen in Figure 2, north of the equator. Given that this stratiform-convective partitioning is relied upon throughout the manuscript, additional analysis and plots would strengthen this study significantly. For example, providing plan view of the convective/stratiform partitioning and radar reflectivity within the domain may alleviate some concerns about the partitioning methods.

6) P10, L8: What is the longitude and time of the cross sections shown in Figure 2? And how do these cross-sections relate to one another? Are they at the same times in the simulations? Are we comparing the same stage of storm development, and are we comparing similar regions within the storm? How were these cross-sections chosen? Without this information, these C/S could be arbitrarily chosen rendering such comparisons meaningless.

7) P10, L19-20: "… the total cloud mass transported into the anvil regions is reduced and the stratiform height is lowered." Is this conclusion being drawn from the C/S? One cannot refer to the total cloud mass when only looking at a C/S. This statement would be better supported by integrating over the anvil area and depths and comparing times series of these masses. Only then is such a statement supported by the output.

8) P10, L24-25: The clouds occurring at the same model time in the same model region with similar SST values does not rule out that differences in dynamics may be impacting differences in cloud structure. For example, what about changes to low-level moisture and cold pool development? This is one of many instances throughout this document where the authors make statements about relative roles of dynamics and microphysics without providing a complete explanation and/or justification of their statement. A similar instance occurs on P10, L28-33. While such statements may be correct, no evidence is provided that clearly supports these comments.

9) P11, L4-5: The authors separate the simulation into periods of strong convection (Hr 10-20) versus weak convection (Hr 21-30). It is unclear how this partitioning was conducted, especially since the maximum vertical velocity continues to be very intense through Hr 23 (Fig 3f). A great deal of analysis throughout the document is based on this partitioning, so additional analysis and/or discussion should be provided to demonstrate that this partitioning is both objectively determined and representative in all cases. Plan views of maximum vertical velocity or reflectivity in the inner domain during both the weak and strong convection times may be useful in helping the reader understand the differences in these regimes. This would also make subsequent discussion of these weak and strong convection regimes much clearer.

10) P11, L5-6: Increasing dust concentrations does not seem to depict lowering cloud field in the D.12 case (Figure 3b). Can you further clarify this statement? Also, it seems like the maximum updrafts in D1.2 are weaker than those in D.12 between 18 and 24 hours, rather than stronger (although the figures are very small so it is difficult to clearly compare the times). Can the authors please comment?

11) P11, L9-10: "The corresponding changes in updraft speed for this period are …" What do these values relate to? This is just one number and yet the plots provided are time series of maximum updraft speed. Please clarify.

12) P11, L15-15: "To illustrate this, … case". This statement needs further clarification. Heterogeneous ice number is not very similar between the D1.2 and D12 cases, as shown in Table 3.

13) Can the authors specify what is meant by heterogeneous and homogenous ice, drop, and graupel number in Table 3?

14) P12, Line 17: The authors state that lower cloud drop numbers are due to increased heterogeneous freezing. Have the authors considered impacts of riming in the different simulations? What role does the collision of liquid droplets with other hydrometeors have in reducing the number of liquid droplets in the simulations?

15) P12, Line 33: The authors state the IN have a minimal impact on liquid nucleation here, but it would be beneficial to remind the readers here (or in the conclusions) that dust can serve highly effectively as a CCN (see papers by Twohy et al and others) and that the impacts of dust on the ice phase may therefore be overestimated by not allowing for dust to serve as CCN and hence be removed in the warm phase.

16) P13, L1-2: "within the selected temperature range of ice/snow . . ." What are these temperature ranges? Include those temperature thresholds here in the text to assist readers.

17) P13, L1-21: Throughout this section of the text, the authors state a lot of microphysical pathways without fully explaining and demonstrating these pathways in the simulations, as discussed in more detail in the next several comments. The authors need to provide a better explanation of the physical processes at play. It is also stated earlier in the manuscript that some of the processes are tracked. The whole of this section would be improved by providing actual magnitudes and/or rates of the physical processes involved.

18) P13, L5-7: Initially, there is an increase in smaller ice crystals in all 3 dust cases, but then there is a decrease in smaller ice crystals in D.12 and D1.2, but not D12. The authors state that this is due to a change in regime of having more particle collisions, as opposed to formation of new ice crystals, but if there was a regime change, then shouldn't this trend also be present in the D12 case? The D12 case continues to have increases in small ice crystal number throughout the simulation. Could the simulation be running out of IN or the convection be cut off from the source of IN during the more intense convective period for the D.12 and D.12 cases? This seems like another plausible explanation. Please will the authors address this in the text?

19) P13, L7-8: The authors attribute the increase in larger liquid drop sizes due to condensation, but what about melting of larger ice crystals that are also present?

20) P13, L9-10: The authors state that the increase in smaller liquid drops is due

to transport from below the freezing level, but this trend is not evident in all the dust simulations and at all the times. Can the authors provide more specifics here?

21) P13, L25-26: What is this ratio in the simulations?

22) P13, L30-34: I do not follow how these methods rule out dynamical effects on the system. This comment is in-line with one of my previous comments in this reagrd. Can the authors provide more explicit discussion here?

23) P14, L5: Can the authors show the stratiform region PSD averages, as well? All of your other plots have 4 columns, and thus it would seem that another column could be added to this plot too. The difference plot of the stratiform region of the Dust case from the Clean case is somewhat meaningless without showing the Clean case PSDs in Figure 6. It is very interesting that the simulations have similar structured PSD in the convective and stratiform region, given the differences in microphysical processes that typically occur within these regions.

24) P14, L11: "although larger sized particles are more slightly common above the freezing level". I am having a hard time seeing this in Figure 6. Which hydrometeor are the authors referring to with the statement?

25) P14, L21-22: Can the authors more explicitly explain how looking at number concentrations before and after the effects of gravity allows for microphysical processes to be tracked? The authors mention instantaneous versus accumulated number on L23-24, but do not provide enough details for the reader to understand what Figure 7 is actually showing. For instance, what time interval is this number flux being calculated over for instance? It takes time for the effects of gravity to operate and for the hydrometeors to fall. This may well be a useful figure and analysis, but it needs to be better described.

26) P15, L25-26: While Saharan dust may undergo less atmospheric processing than Asian dust, it still can serve effectively as CCN, as shown by Twohy et al (2009), the

paper referenced by the authors in the previous paragraph. It might be more beneficial to write this paragraph in terms of representing the thresholds of dust activation as IN and then at CCN and IN, where the studies presented up to this point represent the lowest end of CCN activation.

27) P15, L26-28: The authors mention that prior figures show that the simulations reproduce cloud effects witnessed in a suite of observational studies. Can the authors specify which cloud effects they are referring to (i.e., lower cloud top heights in dusty conditions)? Also, this statement does raise an interesting question regarding the utility of the model. If it is known that dust serve as both CCN and IN, and yet the model output compares well with the observations when dust is only allowed to serve only as IN, how well is the new parameterization actually performing? One way in which to address this would be using measurements of the CCN and/or IN activation capabilities of dust during this field campaign. Either way, a comment should be made in this regard in the manuscript.

28) P16, L19: Should "ice to liquid" be "liquid to ice"?

29) P16, L17-19: "The smaller liquid number at temperatures above freezing can be explained by the reduced ice content immediately above the melting level." Can the authors provide an additional sentence explaining the microphysical process that is causing the smaller liquid number here?

30) P16, L33: From Figure 8c, the D12 case looks to be most different from the other dust cases, not D1.2b as stated in the text. If the authors are only comparing the D1.2a-c cases with this statement, then the authors need to specify this?

31) P17, L3-4: The authors use domain-averaged OLR, but can the authors show OLR only from convective/stratiform columns? Given that cloud occurrence is only ~5% of domain (Figure 8c), it is difficult to justify that the domain-averaged values are represented of the changes to the cloud system.

Figures and Tables

Most of the figures require improvement. The fonts are typically too small, and the sizes of each panel need to be increased. Line thicknesses also should be increased.

1) Figure 1: The color schemes don't seem to match in the two images, despite the fact that the images use the same colorbar. Can you please describe why there is this discrepancy?

2) Figure 3: What are the lower lines in figure 3f, g and h? Are these the strongest downdrafts? This should be made clear in the caption. Also, are these counts of updraft grid points for the whole time period shown in the other figures?

3) Captions are needed for all of the tables in the text.

4) Figure 4: subfigure letters do not match subfigure letters described in text.

5) Figure 5: the titles of the second and third rows do not match the caption. Please correct.

6) Figures 7 and 9: The lines need to be made more distinguishable and thicker. This is especially the case for Figure 9a.

---

## Author Comment (AC1) · 25 Oct 2016

General Comments: This paper investigates the impact of dust acting as IN on tropical convection – specifically its impacts on ice nucleation and particle size distributions -using numerical simulations. The impacts of dust on tropical convection are still not well understood and studies such as this one are needed. Some aspects of the analysis and discussion need clarification, particularly regarding Figure 7 and some of the physical reasoning for the differences seen in the simulations. These are detailed below.

**We thank the reviewer for very thorough and constructive comments. The quality of the manuscript has been improved by these comments and suggestions. Below are our responses (in *Bold)* to the comments. Page and line numbers refer to the original manuscript currently under discussion.**

Specific Comments: 1. Page 2, Line 10: It is confusing to state the increased condensation results from greater droplet nucleation since those are two separate processes. More accurately, the higher droplet concentrations induce the greater condensation and heat release.

**We have clarified the statement on P2, L10 to read: "The higher droplet concentrations induce the greater condensation and latent heat release, resulting in stronger convective updrafts leading to the formation of taller and wider clouds"**

2. Page 5, Lines 12-17: Hasn't the additional IN prognostic variable been available in the SBM for a number of years now?

**We have clarified the statement on P5,L12-17 to acknowledge that the prognostic IN variable has been added in connection with newly implemented set of heterogeneous ice formation mechanisms: "In order to examine IN impacts on clouds and precipitation, an additional prognostic variable for IN particle (dust in this case) number concentration was added to connect our newly implemented heterogeneous ice formation mechanisms with the presence of dust particles in the atmosphere"**

3. Page 6, Line 10: Why is the simplest homogeneous nucleation option being used? Presumably the other options represent homogeneous nucleation more accurately.

**We have replaced the original statement: "The current results use the $H_{thr}$ threshold based scheme to provide pure drop freezing." with**
**"The current results use the $H_{thr}$ threshold based scheme to provide pure drop freezing, which is similar to a number of global climate models. In future studies, we intend to conduct more extensive sensitivity tests related to the different homogeneous freezing mechanisms in conjunction with different partitioning of IN and CCN in the dust layer."**

**The Heymsfield and Milosevich (1993) scheme was intended for use in future related studies, but has not yet been extensively tested within the current model setup. Since this scheme not being used in the current simulation, we have removed the references to the Heymsfield and Milosevich (1993) scheme from Section 2.2.1 (P6,L2-11).**

**In addition, there is an omission in the paragraph relating to the Liu and Penner (2005) aerosol freezing mechanism. The text does not explicitly state that the scheme is not active in the current simulation due to the study focusing on DCC rather than cirrus cloud formation. Since the scheme does not contribute to the results, we have removed the description of this scheme (P6,L13-17)**

4. Page 8, Line 20: If the authors feel that our ability to represent ice nucleation in models is limited and poor, how do we know that the results of this study are meaningful and applicable to the real world?

**We have added an additional clarifying statement to the end of the paragraph on P8, L20+: "Comparison between simulation results and prior observations of our case study will be used to evaluate the model parameterizations implemented into the model for this study."**

5. Page 10, Lines 20-25: I found this part to be confusing. Are the authors simply trying to state that differences in simulated cloud properties are entirely due to dust impacts? I think it is a given that the environment and sea surface are initially the same and therefore do no contribute to differences in the clouds.

**To clarify this, we have replaced the statement: "However, the convective cores presented in Fig. 2 are nearly identically located geospatially, occur at the same model time, and possess nearly identical SST values below the cores, limiting the dynamical effects on the clouds in this instance." with:**

**"However, as all cases are driven by the same initial and boundary conditions, changes to cloud properties noted in Fig. 3 are predominantly affected by changes to microphysical processes, rather than being exclusively a result of differing large scale dynamical conditions."**

**The statements (P10, L20-25) were intended to emphasize that changes to cloud properties were predominantly a result of microphysical processes rather than being exclusively a result of differing large scale dynamical conditions. However, as SST values are fixed in the simulations, this section is not useful for our discussion. Therefore we have removed these lines (P10, L20-25)**

6. Page 11, Line 9: Max updraft speed or mean updraft speed?

**It is the averaged maximum updraft velocity over two periods: a) Hour 10-20, a relatively strong convection period; and 2) Hour 20+, a relatively weak convection period).**

7. Page 11, Lines 14-to end of page: The logic here is circular. First the authors state that the weaker updrafts limit ice nucleation (also, why is this the case?), but aren't the changes in nucleation ultimately the reason for the weaker updrafts?

**Large-scale or environmental dynamics impacts the cloud formation and structure and dust loading also impacts those microphysically through CCN activation and heterogeneous nucleation. We wanted to illustrate both dust IN microphysical effect and environmental dynamical impact based on two different large-scale dynamical strength periods and different dust IN loadings. In the revised paper, we clarified this as**

**"In general, the simulation can be separated into periods of strong (hour 10-20) and weaker (hour 21+) convective activity due to the differing large scale dynamical conditions during these times.**

**After hour 21, the dust cases feature reduced IN activation and domain averaged updraft intensities no longer exceed the Clean case average when compared to the earlier hour 10-20 time range."**
**…**
**"The generally weaker environmental dynamical activity during this period limit heterogeneous ice formation which in turn reduces the contribution of latent heating to parcel buoyancy. To illustrate how the dust effects are limited by both IN availability and by environmental dynamics, we note the similar heterogeneous ice number and updraft changes in the D1.2 (strong convection period) and D12 (weak convection period) cases. During these two time periods, the dust effects are limited either by IN availability (D1.2) or by number of IN activated (D12) yielding similar increases to heterogeneous ice formation and updraft intensity compared to the Clean case."**
**…**

8. Page 12, Line 12: What is meant by "three per unit magnitude increase"**?**

**We have changed the P12, L12 "three per unit magnitude increase" to: "Graupel formation is enhanced compared to the Clean case, with graupel number in the Dust cases increasing by approximately a factor of three per order of magnitude increase of IN concentration from the D.12 case's concentration."**

9. Page 13, Line 13: What is meant be "increases up to 30%, proportionally to IN number"?

**We have changed statement from: "increases up to 30%, proportionally to IN number" to " . . . increases in the dust cases between 5-30% higher than the Clean case, proportionally with increasing IN number . . . "**

10. Page 13, Line 25-26: Does this ratio appear in your simulations?

**We have included the following statement after the sentence ending on P12, L28 to clarify our model agreement: "Our own simulations possess a similar convective/stratiform ratio (~1:6.5) to the ratio reported by Liu and Fu (2001) when averaged over the simulation time to remove the variation resulting from different stages of convective development."**

11. Page 14 and Figure 7: I don't understand what is shown in Figure 7. The authors state it is the difference in location before and after gravitational sedimentation. Are both of these fields output by the model? The titles on the figures say "number flux" but the units are 1/L which is not a flux. Lastly, the text suggests that the authors examine this quantity in order to understand how particles are transported, but are the particles not also transported by the winds? And thus this figure does not really tell us where the regions of formation are?

**Due to the ambiguities related to this figure with respect to the relative contributions of changes to particle terminal velocities and vertical motion in the dust cases, we have replaced Figure 7 with a new figure of calculated fall rates averaged over the convective or stratiform regimes and their corresponding dust case minus clean case differences. This figure is now Figure 8 in the overall order due to the original Figure 9 (and accompanying text) having been moved earlier in the analysis to be Figure 2.**

**In addition, the original passage related to Figure 7 (P14, L19 to P15, L2) has been replaced by the following passages, with summary of key points at the beginning of each paragraph:**

>>Section addresses that cloud geometry is affected by both dynamical and microphysical processes. Describes Figure 8 and specifies how particle fall speed was determined for the figure.

"Differing cloud geometry in the dust cases is a result of changes to the feedbacks between microphysical and dynamical processes within the cloud during formation and growth. Large scale environmental dynamics will provide the baseline values of a cloud's top height, anvil extent, and lifetime (Futyan and Del Genio, 2007), but aerosol indirect effects will modulate these values up or down depending on the changes to the clouds' microphysical processes (Fan et al. 2007a; Koren et al., 2010b; Li Z et al., 2011; Niu and Li, 2012; Fan et al., 2013; Saleeby et al., 2016), especially with respect to changes in hydrometeor PSDs which in turn affect particle terminal velocities (Fan et al., 2013). As previously noted in Figure 4, cloud top height is lowered in our dust simulations despite the presence of increased updraft velocities over the majority of the simulation's time range. In order to further explore this apparent contradiction, Figure 8 provides the convective (row 1 & 2) and stratiform (row 3 & 4) averaged particle fall rates (cm s$^{-1}$) for cloud ice (column 1), snow (column 2), and graupel (column 3) particles, averaged over the total simulation time. Dust case minus Clean case differences for convective and stratiform data are presented in row 2 and row 4, respectively. Particle fall rate is determined by combining calculated particle terminal velocities (positive downwards, Khain and Sednev, 1995) with vertical velocity (positive upwards). In Fig. 8 (row 1 and 3), positive numbers indicate motion towards the surface, while negative numbers indicate the opposite."

>>Addresses changes to cloud ice fall speed and terminal velocity for the different cases at different altitude ranges. Cloud ice is generally heavier (greater terminal velocity) in the dust cases but is more strongly affected by vertical motion.

"Due to their small sizes, cloud ice fall rate is most affected by changes in vertical motion. The negative values of cloud ice fall rate in the convective average (Fig. 8 row 1a) at altitudes between 3 and 13 km indicate that transport of these small particles is predominantly upwards within the convective updrafts. The associated difference plot (Fig. 8 row 2a) indicate that fall rates between 3 and 9 km are increased for the primary dust cases (D.12; D1.2a; D12), and occur in conjunction with the greater updraft speeds at these altitudes. Cloud ice terminal velocities increase by approximately 0.4 (D.12), 1.7 (D1.2a) and 2.6 (D12) cm s$^{-1}$ between 3-9 km, which signify that the ice particles are becoming heavier due to increased diffusional growth.  Above 9 km the fall rates are increased in the D1.2a and D12 cases due to stronger downdraft intensities, as cloud ice terminal velocities are actually reduced by as much as ~1 cm s$^{-1}$ in the dust cases. In the D.12 case cloud ice fall rate and terminal velocities are reduced above 9km when compared with the clean case, while homogeneous freezing is reduced (Table 3). It is reduced by less than one percent in the D.12 case, indicating that many drops are still being transported to temperatures below -38°C and being frozen. While small drop sizes are not necessarily more common in the D.12 case (Fig. 6f), they are also not as significantly reduced as in the D1.2a (Fig. 6g) and D12 (Fig. 6h) cases, leading to a greater number of small ice forming homogeneously near the cloud top and reducing the average cloud ice fall rate. A similar reduction in cloud ice fall rate above 9km is seen in D1.2c. The added CCN in the D1.2c case increases the midlevel liquid content and results in a slightly higher homogeneous freezing number than the base D1.2a case (Table 3). Terminal velocity of the resulting homogeneous cloud ice particles is also slightly reduced compared to the D1.2a case due to the generally smaller sizes of the frozen liquid drops. At altitudes between 3 and 9 km, terminal velocities are nearly identical in both D1.2a and D1.2c cases, with the D1.2c being slightly higher due to increased particle growth. The noticeable difference in fall rate between 6 and 9 km for the D1.2a and D1.2c cases is a result of stronger updrafts in the D1.2c case due to the greater latent heat release from the higher condensate mass. When deposition freezing is removed from the simulation (D1.2b), midlevel ice formation is provided by the immersion freezing mechanism. As

this mechanism freezes liquid drops from the largest sizes to the smallest sizes, the larger drops freeze into graupel rather than cloud ice. Terminal velocities of cloud ice in the D1.2b case is reduced 1-3 cm s$^{-1}$ between 3 and 9 km, which indicates that the large change in fall rate is due to changes in the latent heat profiles affecting vertical motion."

>>Specifies changes to fall rates for snow and graupel. Snow and graupel fall rates are generally reduced between 5 and 10 km and increased above 10km for the primary dust cases. This helps to explain the lowered cloud tops and greater midlevel cloudiness.

"Changes in fall rates of precipitation sizes particles such as snow and graupel will strongly affect eventual surface precipitation accumulation due to changes in downdraft and below-cloud particle residence times and subsequent evaporation. Snow particles tend to grow larger by aggregation processes at warmer temperatures due to greater "stickiness" (Hallgren and Hosler, 1960), which results in fall rates generally increasing towards the surface (Fig. 8 row1b and row 3b). The greater midlevel ice formation in the dust cases results in increased aggregation rates in the 0°C to -38°C temperature range. Larger sizes settle out more quickly and tend to accumulate around the melting level as can be seen between 2 and 5 km. Terminal velocities are increased +15 (D.12) to +60 (D12) cm s$^{-1}$ over this range and are partially countered by stronger updrafts as can be seen in the fall rate differences (Fig. 8 row 2b). At altitudes between 5 and 10 km, snow terminal velocities are decreased between -5 (D.12) and -15 (D12) cm s$^{-1}$ due to more numerous but smaller aggregates, while the resulting fall rates vary between +2 (D.12) to -2 (D12) cm s$^{-1}$ due to the stronger vertical motion in these cases. Above 10 km, terminal velocity and fall rates are increased for the primary dust cases, although D.12 is reduced. In the D.12 case. the most significant ice formation and subsequent aggregation occurs primarily at homogeneous temperatures, yielding smaller aggregates near the cloud top. In the stratiform regime, where vertical motion is weaker, changes in terminal velocity are similar to the convective regime, but are higher overall. This is a result of more active aggregation in the stratiform regime due to the relative lack of liquid water content (for riming) compared to the convective core. When liquid content is significant, graupel forms either by direct freezing of large drops or by riming of existing ice and snow particles. In the primary dust cases (D.12, D1.2a, and D12), stronger updrafts and smaller graupel particles result from the greater midlevel ice concentrations. The graupel fall rates are progressively reduced between 3 and 10 km in both the convective and stratiform regimes as IN concentration is increased. In the D1.2b case, where immersion freezing results in significant formation of graupel from large frozen drops, graupel fall rates are significantly increased. This results in a final accumulated surface precipitation value in the D1.2b case which is 3.7% higher than the Clean case. In contrast, the final values of surface precipitation accumulation are reduced in the primary dust cases (D.12; D1.2a; D12), with the greatest reduction being -6.02% in the D12 case."

12. The authors distinguish between heterogeneous and homogeneous ice throughout the paper. This is based just on the air temperature? But it is possible for heterogeneously nucleated ice to be transported higher in the atmosphere where homogeneous nucleation is dominant, yes? And likewise homogeneously nucleated ice can fall to lower levels. It seems to me that the two types of nucleated ice can't be easily distinguished and that the labels are perhaps misleading.

Yes, heterogeneous and homogeneous regimes are based on the air temperature, and ice particles formed from two different regimes/temperature ranges are transported by the wind. The variables for ice nucleation rates (Fig. 2; Table 3) are not transported by the wind field or gravitational settling, which allows us to determine the location and number of initial ice formation and any relative changes due to dust effects.

**To account for the fact that ice may be transported after formation, the statement of P12, L8-10 has been changed from: "A comparison between the ice number concentration, vertical motion, and CTH for the two convective periods is provided in Table 3. Changes in liquid drop and graupel number have also been provided in Table 3 for the heterogeneous temperature range. " to**

**"A comparison between the dust case minus Clean case hydrometeor number concentration, vertical motion, and CTH for the two convective periods is provided in Table 4. Hydrometeor number concentrations are averaged over the specified temperature ranges. We note that these averages do not directly account for particle transport between different temperature ranges, but rather indicate more generally how the vertical profile of the different hydrometeors are being affected by the different IN concentrations."**

**We have retitled "homogeneous ice number" to "T <-38ºC ice number" and "heterogeneous . . . number" to -38ºC < T < 0ºC . . . number" in Table 4 to clarify that these are temperature based averages.**

13. In Figures 5 and 6, I understand why showing values on a log scale is useful, but I don't understand why the authors add 10 – this just makes the values more difficult to interpret. Also, it is worth pointing out that in the difference of two log10 values is the log10 value of the ratio, i.e. $\log(x)-\log(y)=\log(x/y)$, in order to give more physical meaning to these plots.

**We have removed the scaling factor in Fig. 5 and Fig. 6 to allow the contours to directly represent the $\log_{10}$ values and have clarified that the differences represent the $\log_{10}$ values of the Dust/Clean ratio.**

14. The changes in relative importance of nucleation mechanisms is interesting. I would suggest moving this discussion to earlier in the results section. Changes in nucleation is the first step in the chain of events that lead to the changes in cloud properties, so it seems natural to include this discussion first rather than last in the results section.

**Thank you for your suggestion. We have moved the relevant sections related to initial ice nucleation from section 4.4 to section 4.1 and renumbered the Figures and Tables in the text with the new order.**

15. There have been several studies examining the impacts of dust on tropical convection, particularly in hurricanes, yet in general only those studies by Min et al. are cited. Better citation of other relevant literature is needed.

**We have expanded our references to account for other relevant studies, including:**
**MCS/TC studies: Dunion and Velden (2004), Evan et al (2006), Lau et al (2009), Zhang et al (2009), Braun (2010), Carrio and Cotton (2011), Cotton et al (2012), Storer and Van Den Heever (2013) and Storer et al (2014)**

**Cold pool effects: Altaratz et al. 2007, Berg et al. 2008, Storer et al 2010, Lim et al. 2011, May et al. 2011, Morrison 2012, Grant and Van den Heever, 2015**

**Observations of convective invigoration: Koren et al (2005, 2009); Wall et al (2014)**

16. The lower cloud top heights despite stronger updrafts is a bit confusing, though the authors do give some reasons. I'm wondering if the cloud tops are lower only in the stratiform regions, whereas the strongest updrafts are of course in the convective cores? Perhaps the cloud tops in the convective cores are more similar?

**We have clarified P10, L20: "... stratiform height is lowered" to "Stratiform height is also lowered, despite a similar convective core height. This is consistent with the findings of Min and Li (2010) which described higher convective core heights, shown in their Figure 2, but a lowering of the cloud top heights overall."**

**We have also changed P10,L27"... lowered cloud top height ..." to "lowered overall cloud top height . . ."**

---

## Author Comment (AC2) · 25 Oct 2016

General Comments:
The manuscript addresses several interesting hypotheses based on observations from a group of papers [Min et al. (2009), Min and Li (2010), and Li and Min (2010)] regarding the impacts of dust particles on the characteristics of tropical deep convection via nucleation pathways. Using a suite of simulations that vary the dust concentrations and the hydrometeor activation pathways for one specific case from these observational papers, the authors assess the impact of dust aerosol on ice and cloud nucleation, and subsequent microphysics. The topic of this paper makes it suitable for ACP, and the use of spectral bin microphysics make it unique, however, there are numerous shortfalls in the manuscript that currently make it unsuitable for publication in ACP, including (1) a lack of analysis and/or justification for a number of the statements made; (2) the absence of number relevant citations as well as placing the current research within the context of past work; (3) little to no demonstration regarding how well the case study output compares with the observations; (4) issues related to the convective-stratiform partitioning and the convective thresholds; (5) little to no substantiated physical reasoning in the role of the dynamics; and (6) poor figure quality. These are outlined in detail below.

**We greatly appreciate the reviewer's time spent reading and making comments to improve our manuscript. We will address each comment point by point below. Our responses and text modifications will appear in bold. Line numbers refer to the original manuscript currently under discussion.**

**We briefly summarize what we have done to address the major comments that the reviewer listed here. In order to improve our analysis, we have expanded our descriptions of microphysical processes and added specific values for rates or other such changes. We have included additional citations where these were suggested or were otherwise appropriate. We have improved figure quality in order to increase visibility. In addition, we have replaced the figure related to particle sedimentation with a new figure that takes into account changes to both particle terminal velocities and vertical motion. We have clarified the reasoning behind our choice of convective/stratiform classification and its correspondence with observed convective/stratiform ratios. We have also clarified the individual roles of dynamical and microphysical processes in our discussion.**

Introduction and References

1) P1 L27: In addition to convective intensity, aerosols have also been found to influence the anvil characteristics as described in papers by Fan et al (2007) and a recent paper by Saleeby et al (2016).

**We have changed the P1, L26-27 sentence from: "The convective intensity controls the depth, area, and lifetime of the resulting anvil clouds (Futyan and Del Genio, 2007)." to**

**"The convective intensity is the primary determiner of the depth, area, and lifetime of the resulting anvil clouds (Futyan and Del Genio, 2007), but changes to cloud microphysical processes resulting from aerosol indirect effects (AIE) will modulate these qualities differently depending on the aerosol type (Fan et al. 2007a; Min et al., 2009; Koren et al., 2010b; Li Z et al., 2011; Niu and Li, 2012; Fan et al., 2013; Saleeby et al., 2016)."**

2) P2 L4-19: The authors have cited a lot of their own work, which is fine, however, I would encourage them to cite other relevant work here too. For example, there are numerous other observational and modeling papers examining the impacts of Saharan dust on deep convective systems including MCSs and hurricanes that are relevant to this work that have not been cited including Dunion and Velden (2004), Evan et al (2006), Lau et al (2009), Zhang et al (2009), Braun (2010), Carrio and Cotton (2011), Cotton et al (2012), Storer and Van Den Heever (2013) and Storer et al (2014). Some of these references are also relevant to the statements made on page 3.

**We have incorporated these additional references as suggested into the relevant locations of P3, L2-3**

3) P2 L12: There are numerous observational and modeling papers that have shown that cold pools are warmer in more polluted environments (Altaratz et al. 2007, Berg et al. 2008, Storer et al 2010, Lim et al. 2011, May et al. 2011, Morrison 2012, Grant and Van den Heever, 2015). These papers argue that raindrops are larger under more polluted conditions due to greater rain accretion of the higher cloud water contents produced in more polluted conditions. Please be sure to include this argument too in your description of cold pool impacts.

**We have incorporated the following to P2,L13: "Conversely, other studies have noted that the formation of larger drops due to enhanced rain drop accretion processes will limit evaporation and weaken the cold pool (Altaratz et al. 2007; Berg et al. 2008; Lerach et al., 2008, Storer et al., 2010; Lim et al., 2011; May et al., 2011, Morrison, 2012; Grant and Van Den Heever, 2015). "**

4) P2 L21: It should be notes that dust can serve as CCN and/or IN as shown by various papers by Twohy et al (2009 and others)

**We have expanded P2, L33 from: ". . . have been established to act as effective IN (Pruppacher and Klett, 1997; Demott et al., 2003; Sassen et al., 2003)" to**

**". . . Sassen et al., 2003) and/or CCN (Twohy et al., 2009; Kumar et al., 2011; Karydis et al., 2013)."**

5) P2 L23: The paper by Van den Heever et al (2006) which the authors cite earlier, was one of the first to investigate the impacts of both CCN and IN on DCCs and should be cited here.

**In order to acknowledge the Van den Heever (paper), we have added a brief discussion of Van den Heever et al. (2006) immediately prior to the Ekman et al. (2007) and Fan et al. (2010) discussions in the after the sentence ending P3, L3:**

**"The study of Van den Heever et al. (2006) described the differing impacts of CCN and IN on convective clouds and subsequent anvil development. They found that increasing CCN concentration tended to reduce surface precipitation, while increased IN concentration initially increased surface precipitation, but reduced the total to less than the clean simulation by the end of the simulation. Updraft intensity increased with the increased aerosol concentration due to stronger latent heat release, but anvils were generally smaller and more organized."**

6) P2 L26: While these are all fine references cited here, these papers did not "discover" heterogeneous nucleation per se. The authors should go back to the original statements on these processes, or alternatively state what the papers referenced here added to the field of heterogeneous nucleation.

**In the statement on P2, L25-28, we have cited (Vali et al., 1985) in defining the terminology used in this section related to Heterogeneous freezing and have clarified the previously mentioned references' contributions.**

7) P3 L24-25: The past research of Koren et al (2005, 2009) and Wall et al (2014) also utilized observations to examine convective invigoration in response to dust across the globe.

**We have expanded our P3,L24 statement to account for other observations that address convective invigoration in connection with aerosol indirect effects: (Koren et al., 2005,2010; Min et al., 2014; Storer et al., 2014; Wall et al., 2014)**

Experiment Setup

1) P5 L12-17: "sources of dust (from lateral boundaries)" – are the lateral boundaries the only sources of dust in this model setup? Please clarify. Also, it is stated that "such a layer has been added". How was this later added? What are the characteristics of this layer? Finally, how is the fractional combination of dust serving as CCN and IN parameterized? And what was it set to in these simulations?

**We have added the following to P5,L14: "The dust layer is initialized to cover the entire domain at model startup and thereafter is supplied exclusively from the lateral boundaries by wind advection."**

**And P5,L16: "The dust in the layer can serve as IN, CCN, or some fractional combination of the two by means of a simple partition which is set by the user depending on assumed or measured particle chemistry."**

2) P6, L10: Why did the authors choose to use the Hthr homogenous nucleation method over the other two methods described? Furthermore, what is the point of describing the H&M method if it is simply an addition to the model that was not utilized in this

**We have replaced the original statement (P6, L10-11): "The current results use the $H_{thr}$ threshold based scheme to provide pure drop freezing." with "The current results use the $H_{thr}$ threshold based scheme to provide pure drop freezing, which is similar to a number of global climate models. In future studies, we intend to conduct more extensive sensitivity tests related to the different homogeneous freezing mechanisms in conjunction with different partitioning of IN and CCN in the dust layer."**

**The Heymfield and Milosevich (1993) scheme was intended for use in future related studies, but has not yet been extensively tested within the current model setup. Since this scheme not being used in the current simulation, we have removed the references to the Heymfield and Milosevich (1993) scheme from Section 2.2.1 (P6,L2-11).**

**In addition, there is an omission in the paragraph relating to the Liu and Penner (2005) aerosol freezing mechanism. The text does not explicitly state that the scheme is not active in the current simulation due to the study focusing on DCC rather than cirrus cloud formation. Since the scheme does not contribute to the results, we have removed the description of this scheme (P6,L13-17).**

3) P6 L20-21 (and later P8 L20): The authors state "Deposition nucleation should be important for deep convection, but there is no such a scheme developed for deep convection with a connection with aerosols." They then go on to state in the same paragraph "To link depositional / condensational freezing with aerosols, we follow the implementation of van den Heever et al (2006), update from the Meyers et al (1992) version." It would thus seem that such a scheme has been developed before. Furthermore, from what I recall, such a scheme does exist in the RAMS model used in the Van den Heever study. Do the authors mean that no such scheme exists in this version of the SBM? Please clarify this in the text.

**We have clarified the sentence as "Currently there is no deposition nucleation parameterization connecting with aerosol properties and developed based on deep convective clouds". Indeed Van den Heever (2006) connected Meyers et al. 1992 with aerosols but it is not the ice nucleation parameterization originally developed based on deep convective cloud data.**

**To our knowledge, all the recent parameterizations that connect with aerosols were developed from laboratory and field experiments from other cloud types, but not deep convection.**

4) P6, L24: Please specify temperature range.

**We have added the clarification to P6, L24 from: "in the temperature range" to "in the 0°C to -10°C temperature range"**

5) P6-P7, Eq (1-2): Are Nid and NIN the same variable? Can you please clarify?

**We have changed P6,L28 from: "the IN number concentration ($N_{IN}$; $l^{-1}$)" to "the IN number concentration ($N_{IN}$; $l^{-1}$) within the grid cell" and P7, L2 from "fraction of available IN (Nid; $l^{-1}$)" to "fraction of the maximum available IN (Nid; $l^{-1}$) concentration that may be activated for the given conditions."**

6) P7-8, Eq (4-7): Is the explanation of how NCNT is calculated necessary if the authors are just keeping it set at a constant value?

**Since the mechanism was updated from the original version to account for the SBM's drop bin distribution and prognostic IN variable, we felt it was useful to explicitly state the equation. NCNT does vary with temperature, as well as number and size of both drops and IN and so is not strictly constant even in a specific drop bin.**

7) P8, Eq (7): State what "r" stands for in Eq 7.

**"r" is the average dry aerosol radius.**

**We have added this clarification to P8, L5.**

8) P8, L27: What is the grid spacing for the 41 vertical levels in the simulations? Also, 3km and 41 vertical levels seems relatively coarse for resolving the updrafts and cloud processes of MCSs. Have the authors tested the sensitivity to grid resolution in any way? This could be tested with a less costly bulk scheme to ensure that enhancing grid resolution doesn't have a significant impact on the case study results.

**We have added a description of the vertical grid resolution to P8,L27.**

**The grid spacing is coarsest at the top of the model (~500m), becoming finer near the surface. We have not significantly tested the effects of grid resolution on the case study, but do agree that a finer resolution would allow for smaller cloud features to be better resolved. Computational expense is an issue with the SBM at the current time, which limits significant grid resolution sensitivity testing. Our concern with using a bulk model for comparison in a sensitivity test is due to the differences in microphysical calculations between bin and bulk microphysics. Bulk model resolution sensitivity may not be representative of similar sensitivity in the SBM.**

9) P9, L3: While the large scale precipitable water patterns are relatively well produced, the magnitudes are quite different. This could have a significant impact on both the amount of liquid water and ice formed by the model. Please can the authors comment on this?

The available precipitable water in the atmosphere will impact cloud formation processes. What we focus on is the effects of different dust loading, in terms of IN concentration, on cloud and precipitation processes for a given large-scale dynamical field (including precipitable water). The differences between the simulated and observed precipitation water would have small impacts on our conclusion drawn from relative differences under the same large-scale dynamical field in all simulations.

**We have changed the statement of P9, L3 from "The large scale patterns are well reproduced by the model, assuring that meteorological conditions in the region of interest are being represented correctly." to**

**"The large scale patterns are well reproduced by the model, although we note that the magnitude differs over land and the southern Atlantic by ~7 kg m$^{-2}$compared to observations in some areas. Despite this, the magnitude in the location of our smallest domain is well reproduced, suggesting that the meteorological conditions in our region of interest will be represented sufficiently well."**

10) P16, L2: Adding 12 CCN cm-3 to the dust layer for case D1.2c seems like a somewhat arbitrary number, especially given that the numbers measured were more than double this (_30/cc) as stated in the paper. Why not add 30/cc? Also, surely a more correct approach here is not to add more CCN, but rather attribute some portion of the initial dust population to CCN and some to IN, and hence remove the number that are allowed to act as CCN from the number that were allowed to operate as IN in the first set of simulations? In the current approach, the authors have simply added more dust particles to the environment. This does not appear to represent the problem correctly, in that when dust can act as CCN or IN, when they operate as CCN this prevents them from acting as IN (unless through immersion freezing) as they are typically rained out. In the current approach, the simulations are simply being made more

polluted, and hence the effects of greater dust concentrations, as opposed to a different partitioning between CCN and IN, is being investigated. Please will the authors comment on this approach?

**The choice of adding 12pcc of CCN to the D1.2c case was based on 12pcc being the greatest IN number under consideration within our current study. The original AEROSE dust measurement (Morris et al., 2006) of 30pcc was applied to the IN active fraction of Saharan dust suggested by Niemand et al. (2012) and yielded the 0.12pcc value, which was then increased by one or two orders of magnitude for the other dust case sensitivity studies. The addition of 12pcc (or 30pcc) of CCN was assumed to be relatively insignificant with respect to the background values of CCN, but was included to rule out significant effects on the results that may otherwise have been overlooked.**

**The more rigorous partitioning of dust between CCN and IN based on more detailed measurements of particle qualities within the Saharan dust outbreak is intended for future study.**

Case Study Evaluation

1) P9, L3: The authors motivate this study and discuss the results of this study based on a suite of observational studies by Min et al. (2009), Li and Min, (2010), and Min and Li (2010). However, there is not enough analysis presented in this study to demonstrate whether the simulations are properly reproducing the MCS event on 08 March 2004. The only comparison to observations is a precipitable water comparison, which is averaged over a 3-day period. While this does provide some information about the large-scale environment, it does not provide enough information on whether the convective elements and environment are similar to those observed and discussed in the observations studies listed above. Many studies have shown that aerosol effects on convection may depend on the convective system structure and environmental conditions, some of which are cited in this document. Therefore, the authors should include additional comparison of the simulations with observations in terms of the environment and convective system structure, especially since the authors are trying to explain an observed phenomenon represented by these simulations. Given that the observations from this case study have been published by some of the authors, it seems reasonable to compare some of the observations put forth in these prior studies to the same values in the simulations.

**The current study is intended to be an idealized case study focusing on the sensitivity of the observed DCC to IN activation exclusively. Subsequent studies are intended to test the partition of dust between IN and CCN in order to improve the realism of the simulations with respect to observations.**

**The below figure compares the time series of (3-hour) TRMM observed and Clean case model output rain rates averaged over the smallest model domain. The Clean case values are similar, but slightly lower than the observed values with peak precipitation occurring later. Since the dust outbreak occurring during the study period will add additional IN and CCN to the environment, the Clean case represents our idealized baseline for comparison. We have added the below plot to Figure 1 to provide additional comparison between model results and observed values in the location of the smallest domain.**

[Figure]

(c) Smallest Domain Rain Rate

2) P9 line 11: The authors state that dust number concentrations were enhanced within the layer between 1 and 3 km. What did the authors do for the regions not included within the SAL? What were these concentrations set to? The SAL would not realistically extend across the entire region captured by their grid 1. Also, the authors state that these concentrations are those for IN. Can these particles also serve as CCN? The authors refer to a relative combination of serving as CCN and/or IN earlier in the paper, however, that is not indicated here.

**We have added the following statement after P9, L11: "The dust layer contributes IN to the smallest domain only, as the bulk microphysics used in the larger domains do not directly connect dust with ice formation."**

**In the larger domains, CCN values were assumed to typical maritime or continental values based on the grid locations. The current simulations have dust serving exclusively as IN, but one case (D1.2c) incorporated additional CCN to test the sensitivity of dust acting as CCN in addition to IN.**

3) P9 line 12: There seems to be an error in one of these case study names as two of them are the same.

**Case D1.2 was mislabeled as D.12.**

**We have corrected the typo on P9,L12.**

4) P9, L24-26: The SAL is typically a dry layer, with layers that tend to be moister below and above the SAL. Why is the dryness of SAL not represented here? Also, what data are used to provide the initial environmental data, as well as the boundary conditions?

**We have added a statement after P9, L26 addressing the presence of the SAL: "After the model's 6 hour spin up time, a dry air layer corresponding to the SAL enters the domain via the NCEP-FNL boundary conditions and is present for the duration of the simulation."**

Scientific Analysis, Reasoning, Clarity and Conclusions

1) P1 L15: "Ice particle size distributions" – which ice particles are the authors referring to? Smaller ice, larger ice, all of the ice species? As ice can refer collectively to small and large ice species, this should be clarified.

**We have clarified that cloud ice is being referred to in the P1, L15 statement and have added a further clarification after this sentence: "Snow PSD increases at the larger sizes due to more active aggregation processes in the dust cases."**

2) P1 L20: "which is consistent with observations" – which observations are these? Do the authors mean observations of the current case? Or to observations more generally. If the latter, other observations have not always found this to be the case.

**We have clarified P1,L20 from : "…consistent with observations." to "…consistent with observations of the case study [Min and Li, 2010]."**

3) P2 L13-16: There are various definitions of convective invigoration ranging from stronger updrafts, to higher cloud tops, to more precipitation production to impacts on latent heating, with some of these being related. I would recommend that the authors make clear what they mean by convective invigoration, thermodynamic invigoration etc.

**We have clarified the P2,L13-17 statements beginning with: "Aerosol related changes to cloud macrophysics . . . " to:**

**"Aerosol indirect effect related changes to cloud macrophysics are frequently attributed solely to thermodynamic invigoration (stronger latent heat release) induced by the increased liquid and/or ice particle number concentrations and subsequent changes to diffusional growth processes. However, a study by Fan et al. (2013) involving simulations of DCC in three different regions, suggested that the observed taller and wider clouds cloud be better explained by changes to the microphysical properties such as the particle size distribution. The thermodynamic invigoration by increased latent heat release did not unanimously occur in the study when polluted conditions were simulated, although increased cloud fraction and cloud top height were present."**

4) P4 L9: Do note that some bulk schemes diagnose the shape parameter.

**We have clarified the P4, L9-10 statement to read: "Unlike most bulk microphysics schemes, in which the PSD of every predicted hydrometeor type has a predefined shape or is determined by a semi-empirical relationship, bin microphysics schemes allow PSDs to evolve naturally in conjunction with in-cloud and large scale processes."**

5) P10, L1-3: The authors partition precipitating columns into convective and stratiform regions, using somewhat arbitrary thresholds for vertical motions and cloud depth. Furthermore, there seems like there

may be a few loopholes in their classification. For example, if a column has cirrus clouds in the upper troposphere, but precipitating, shallow warm-phase clouds below it, it seems like these would be classified as stratiform. Is this correct? If not, and these are classified as warm clouds, then the upper level cirrus would be excluded from their analysis. An example of such an instance can be seen in Figure 2, north of the equator. Given that this stratiform-convective partitioning is relied upon throughout the manuscript, additional analysis and plots would strengthen this study significantly. For example, providing plan view of the convective/stratiform partitioning and radar reflectivity within the domain may alleviate some concerns about the partitioning methods.

**In the case of multiple cloud layers, classification is top down, which is consistent with satellite based cloud observations that only view the top most cloud layer. In the reviewer's example of cirrus clouds overlaying a precipitating warm cloud, the column would be classified as stratiform.**

**Our current convective stratiform selection criteria are similar to the TRMM based classification of Awaka et al. (1997) for each type of cloud, but accounts for vertical motion and cloud thickness in determining deep convection. The TRMM based classification was used in our original observational studies (Min et al., 2009; Min and Li, 2010; Li and Min, 2010). Applying the current criteria to the model results does match well with TRMM based convective/stratiform ratio (~1:7 compared to the model's ~1:6.5).**

6) P10, L8: What is the longitude and time of the cross sections shown in Figure 2? And how do these cross-sections relate to one another? Are they at the same times in the simulations? Are we comparing the same stage of storm development, and are we comparing similar regions within the storm? How were these cross-sections chosen? Without this information, these C/S could be arbitrarily chosen rendering such comparisons meaningless.

**The selected cross sections are both taken from model hour 15. The slices are not identically located, but are less than 4 grid points apart and are binned zonally over 9km.**

**We have added the clarifying statement to P10,L10: "The slices are not identically located in the two cases due to small differences in the spatial evolution of the system, but are less than 4 grid points apart. In both cases, the slices are similarly located within their respective cloud system and are at similar stages of evolution. The slices are binned zonally over 9km to further reduce the effects of spatial variations." and have indicated that both slices occur at hour 15.**

7) P10, L19-20: ": : : the total cloud mass transported into the anvil regions is reduced and the stratiform height is lowered." Is this conclusion being drawn from the C/S? One cannot refer to the total cloud mass when only looking at a C/S. This statement would be better supported by integrating over the anvil area and depths and comparing times series of these masses. Only then is such a statement supported by the output.

**The conclusion was based on the changes to TWC and cloud mask outline in the stratiform and anvil cloud regions. The figure better illustrates the changes to stratiform cloud height and anvil extent than accurate cloud mass changes. We will change this reference in the text.**

**We have changed the P10,L19 statement from: ". . . the total cloud mass transported into the anvil region is reduced . . . " to "Despite the higher vertical motion evident in the D1.2 case, total anvil extent is reduced. Stratiform height is also lowered, despite a similar convective core height."**

8) P10, L24-25: The clouds occurring at the same model time in the same model region with similar SST values does not rule out that differences in dynamics may be impacting differences in cloud structure. For example, what about changes to lowlevel moisture and cold pool development? This is one of many instances throughout this document where the authors make statements about relative roles of dynamics and microphysics without providing a complete explanation and/or justification of their statement. A similar instance occurs on P10, L28-33. While such statements may be correct, no evidence is provided that clearly supports these comments.

**We have changed the P10,L23 from: "However, the convective cores presented in Fig. 2 are nearly identically located geospatially, occur at the same model time, and possess nearly identical SST values below the cores, limiting the dynamical effects on the clouds in this instance." to:**

**"However, as all cases are driven by the same initial and boundary conditions with fixed SST, changes to cloud properties noted in Fig. 3 are predominantly caused by changes to microphysical processes, rather than being exclusively a result of differing large scale dynamical conditions. Note that the microphysical changes may feedback to the local dynamics such as cold pools or buoyancy that can change cloud properties as well."**

**Also we have removed P10,L28-33: "The study of Min and Li (2010), expanded upon these findings, reporting both a positive correlation between cloud water path and cloud top height and a negative correlation between cloud top height and dust AOD. This provides further support for the hypothesis that the results from the dusty region were not strictly a result of dynamical effects."**

9) P11, L4-5: The authors separate the simulation into periods of strong convection (Hr 10-20) versus weak convection (Hr 21-30). It is unclear how this partitioning was conducted, especially since the maximum vertical velocity continues to be very intense through Hr 23 (Fig 3f). A great deal of analysis throughout the document is based on this partitioning, so additional analysis and/or discussion should be provided to demonstrate that this partitioning is both objectively determined and representative in all cases. Plan views of maximum vertical velocity or reflectivity in the inner domain during both the weak and strong convection times may be useful in helping the reader understand the differences in these regimes. This would also make subsequent discussion of these weak and strong convection regimes much clearer.

**Convective averages of vertical motion show a distinct transition between hour 21 and 22, which can be seen in the plot below. The white line denotes hour 22**

**We have added a clarifying statement to P11,L5: "In general, the simulation can be separated into periods of strong (hour 10-20) and weaker (hour 21+) convective activity due to the differing large scale dynamical conditions during these times. After hour 21, the dust cases feature reduced IN activation and domain averaged updraft intensities no longer exceed the Clean case average when compared to the earlier hour 10-20 time range."**

[Figure]

**Clean Vertical Motion (cm s−1)**

10) P11, L5-6: Increasing dust concentrations does not seem to depict lowering cloud field in the D.12 case (Figure 3b). Can you further clarify this statement? Also, it seems like the maximum updrafts in D1.2 are weaker than those in D.12 between 18 and 24 hours, rather than stronger (although the figures are very small so it is difficult to clearly compare the times). Can the authors please comment?

**In the D.12 case the small IN number limits the effects to subtle changes that may locally increase or decrease the cloud height, but overall results in a degree of lowering as noted in the time averaged value of CTH for the two convective periods (Table 4). The D1.2 updrafts between 18-24 are weaker due to the enhanced latent heat release not overcoming the increased parcel mass from the greater ice nucleation. Averaged updraft condensate mass flux in the D1.2 case during this time is increased by ~44% from the Clean case, while total latent heat release is actually decreased by ~12%.**

11) P11, L9-10: "The corresponding changes in updraft speed for this period are : : :" What do these values relate to? This is just one number and yet the plots provided are time series of maximum updraft speed. Please clarify.

**The single values of maximum updraft speed are time averaged over the duration of the strong or weak convective periods.**

**We have clarified P11,L9 from : "The corresponding changes in updraft speed . . . " to "The corresponding time averaged changes in maximum updraft speed . . . "**

12) P11, L15-15: "To illustrate this, : : : case". This statement needs further clarification. Heterogeneous ice number is not very similar between the D1.2 and D12 cases, as shown in Table 3.

**We have changed the P11,L14-16 statement from: "The generally weaker updrafts during this period limit new ice formation which reduces the contribution of latent heating to parcel buoyancy. To illustrate this, we note the similar heterogeneous ice number and updraft changes in the D1.2 (strong convection) and D12 (weak convection) cases." to:**

**"The generally weaker environmental dynamical activity during this period limit new heterogeneous ice formation which in turn reduces the contribution of latent heating to parcel buoyancy. To illustrate how the dust effects are limited by both IN availability and by environmental dynamics, we note the similar heterogeneous ice number and updraft percent difference changes in the D1.2 (relatively strong convection period) and D12 (relatively weak convection period) cases. During these two time periods, the dust effects are limited either by IN availability (D1.2) or by number of IN activated (D12) yielding similar percent increases to heterogeneous ice formation and updraft intensity compared to the Clean case."**

13) Can the authors specify what is meant by heterogeneous and homogenous ice, drop, and graupel number in Table 3?

**The statement of P12,L8-10 has been changed from: "A comparison between the ice number concentration, vertical motion, and CTH for the two convective periods is provided in Table 3. Changes in liquid drop and graupel number have also been provided in Table 3 for the heterogeneous temperature range. " to**

**"A comparison between the dust case minus Clean case hydrometeor number concentration, vertical motion, and CTH for the two convective periods is provided in Table 4. Hydrometeor number concentrations are averaged over the specified temperature ranges. We note that these averages do not directly account for particle transport between different temperature ranges, but rather indicate more generally how the vertical profile of the different hydrometeors are being affected by the different IN concentrations."**

**We have retitled "homogeneous ice number" to "T <-38ºC ice number" and "heterogeneous . . . number" to -38ºC < T < 0ºC  . . . number" in Table 4 to clarify that these are temperature based averages.**

14) P12, Line 17: The authors state that lower cloud drop numbers are due to increased heterogeneous freezing. Have the authors considered impacts of riming in the different simulations? What role does the collision of liquid droplets with other hydrometeors have in reducing the number of liquid droplets in the simulations?

**The original statement did not clearly express that riming is a significant process in converting liquid to ice in our simulation. We have expanded our statement to specifically address this omission.**

**We have replaced the P12,L17 statement: "Coupled with the similarly progressive increase in graupel number, the loss of liquid drops suggests that fewer drops are reaching the -38°C temperature range to freeze homogeneously in the Dust cases due to the heterogeneous freezing resulting in a much greater conversion of liquid to ice. " with**

**"Coupled with the similarly progressive increase in graupel number, which indicates more frequent riming, the loss of liquid drops in the 0°C to -38°C temperature range suggests that fewer drops are reaching the -38°C temperature range to freeze homogeneously in the dust cases. This is due to increased heterogeneous ice formation resulting in a much greater conversion of liquid to ice, by direct freezing, riming and Bergeron evaporation."**

15) P12, Line 33: The authors state the IN have a minimal impact on liquid nucleation here, but it would be beneficial to remind the readers here (or in the conclusions) that dust can serve highly effectively as a CCN (see papers by Twohy et al and others) and that the impacts of dust on the ice phase may therefore be overestimated by not allowing for dust to serve as CCN and hence be removed in the warm phase.

**We have added a note to the end of P12,L34: "As dust in nature can also act as effective CCN and may therefore be removed from the system by warm rain processes before freezing occurs, these results should be interpreted as an upper range of IN effects for a given number concentration."**

16) P13, L1-2: "within the selected temperature range of ice/snow : : :" What are these temperature ranges? Include those temperature thresholds here in the text to assist readers.

**We have clarified the text to indicate the temperature range is 0C to -38C, corresponding with the heterogeneous freezing regime.**

17) P13, L1-21: Throughout this section of the text, the authors state a lot of microphysical pathways without fully explaining and demonstrating these pathways in the simulations, as discussed in more detail in the next several comments. The authors need to provide a better explanation of the physical processes at play. It is also stated earlier in the manuscript that some of the processes are tracked. The whole of this section would be improved by providing actual magnitudes and/or rates of the physical processes involved.

**In order to improve our discussion of snow and graupel growth, we have expanded this section with additional details relating to the processes of aggregation and riming:**

**After P13,L7 we have added: "The average rate of aggregation between 0C and -38C in the dust cases differs from the Clean case by -10.6% (D.12), +388.6% (D1.2a), and +4.5K% (D12). The large increases in the D1.2a and D12 cases are a result of the larger midlevel ice number in comparison with the Clean and D.12 cases, which allows for more frequent ice particle collisions. The reduced aggregation rate in the D.12 case is a result of reduced collection efficiency due to smaller ice**

**particle sizes, without the significant increase in deposition-condensation activation (Table 3) and heterogeneous ice number concentration (Table 4) that occurs in the higher IN number cases."**

**After the statement ending P13,L21 we have added: "Due to this competition, the actual graupel mass formed by riming processes in the temperature range between 0°C and -38°C decreases from the Clean case average by -22.3% (D.12), -42.3% (D1.2a), and -51.5% (D12), respectively"**

18) P13, L5-7: Initially, there is an increase in smaller ice crystals in all 3 dust cases, but then there is a decrease in smaller ice crystals in D.12 and D1.2, but not D12. The authors state that this is due to a change in regime of having more particle collisions, as opposed to formation of new ice crystals, but if there was a regime change, then shouldn't this trend also be present in the D12 case? The D12 case continues to have increases in small ice crystal number throughout the simulation. Could the simulation be running out of IN or the convection be cut off from the source of IN during the more intense convective period for the D.12 and D.12 cases? This seems like another plausible explanation. Please will the authors address this in the text?

**In all cases there is still a residual supply of IN that remain un-activated (equal to or greater than the background IN concentration) and are transported by the wind field. The regime change is present in the D12 case (Note the hook shape in Fig. 5d around hour 18), but is less obvious due to the greater total ice nucleation resulting from the higher IN number.**

19) P13, L7-8: The authors attribute the increase in larger liquid drop sizes due to condensation, but what about melting of larger ice crystals that are also present?

**The statement P13,L7-8: " The liquid PSD describes enhancement to the largest drops resulting from non-freezing collection of small ice crystals and/or some degree of enhanced condensational growth in the stronger updrafts present. " has been changed to:**

**"The liquid PSD describes enhancement to the largest drops that could be the result of increase collision-coalescence and/or the melting of large ice particles."**

20) P13, L9-10: The authors state that the increase in smaller liquid drops is due to transport from below the freezing level, but this trend is not evident in all the dust simulations and at all the times. Can the authors provide more specifics here?

**It should be specified that the increased transport of small drops only occurs towards the end of the strong convective period and during the weaker convective period, when ice nucleation is lower compared to earlier in the strong convective period. Greater ice contents will increase the frequency of collisions and increase the effects of Bergeron evaporation.**

**The P13,L9-10 statement : "The greater presence of smaller drops is due to increased transport from below the freezing level, as the small sizes limit the efficiency of freezing and collection. " has been changed to:**
**"The greater presence of smaller drops primarily during the weak convective period is due to increased transport from below the freezing level, as the small sizes limit the efficiency of freezing and collection and the smaller ice number concentration limits Bergeron evaporation in comparison with the strong convective period."**

21) P13, L25-26: What is this ratio in the simulations?

**We have added a clarifying statement after the statement ending P13,L28: "Our own simulations possess a similar convective/stratiform ratio (~1:6.5) to the ratio reported by Liu and Fu (2001) when averaged over the simulation time to remove the variation resulting from different stages of convective development."**

22) P13, L30-34: I do not follow how these methods rule out dynamical effects on the system. This comment is in-line with one of my previous comments in this regard. Can the authors provide more explicit discussion here?

**We have deleted that sentence.**

23) P14, L5: Can the authors show the stratiform region PSD averages, as well? All of your other plots have 4 columns, and thus it would seem that another column could be added to this plot too. The difference plot of the stratiform region of the Dust case from the Clean case is somewhat meaningless without showing the Clean case PSDs in Figure 6. It is very interesting that the simulations have similar structured PSD in the convective and stratiform region, given the differences in microphysical processes that typically occur within these regions.

**We have added the Clean case stratiform average to the Figure 6.**

**The similarity in the PSDs for the convective and stratiform regimes between the cases is due to the majority of ice formation and initial growth occurring in the convective core before being transported out into the stratiform regime. Particles that are transported into this regime tend to grow by aggregation rather than riming due to the lack of supercooled drops and therefore yield larger snow particles.**

24) P14, L11: "although larger sized particles are more slightly common above the freezing level". I am having a hard time seeing this in Figure 6. Which hydrometeor are the authors referring to with the statement?

**We have clarified that snow is being referred to in the statement on P14;L11.**

25) P14, L21-22: Can the authors more explicitly explain how looking at number concentrations before and after the effects of gravity allows for microphysical processes to be tracked? The authors mention instantaneous versus accumulated number on L23-24, but do not provide enough details for the reader to understand what Figure 7 is actually showing. For instance, what time interval is this number flux being calculated over for instance? It takes time for the effects of gravity to operate and for the hydrometeors to fall. This may well be a useful figure and analysis, but it needs to be better described.

**Due to the ambiguities related to this figure with respect to the relative contributions of changes to particle terminal velocities and vertical motion in the dust cases, we have replaced Figure 7 with a new figure of calculated fall rates averaged over the convective or stratiform regimes and their corresponding dust case minus clean case differences. This figure is now Figure 8 in the overall**

order due to the original Figure 9 (and accompanying text) having been moved earlier in the analysis to be Figure 2.

In addition, the original passage related to Figure 7 (P14, L19 to P15, L2) has been replaced by the following passages, with summary of key points at the beginning of each paragraph:

>>Section addresses that cloud geometry is affected by both dynamical and microphysical processes. Describes Figure 8 and specifies how particle fall speed was determined for the figure.

"Differing cloud geometry in the dust cases is a result of changes to the feedbacks between microphysical and dynamical processes within the cloud during formation and growth. Large scale environmental dynamics will provide the baseline values of a cloud's top height, anvil extent, and lifetime (Futyan and Del Genio, 2007), but aerosol indirect effects will modulate these values up or down depending on the changes to the clouds' microphysical processes (Fan et al. 2007a; Koren et al., 2010b; Li Z et al., 2011; Niu and Li, 2012; Fan et al., 2013; Saleeby et al., 2016), especially with respect to changes in hydrometeor PSDs which in turn affect particle terminal velocities (Fan et al., 2013). As previously noted in Figure 4, cloud top height is lowered in our dust simulations despite the presence of increased updraft velocities over the majority of the simulation's time range. In order to further explore this apparent contradiction, Figure 8 provides the convective (row 1 & 2) and stratiform (row 3 & 4) averaged particle fall rates (cm s$^{-1}$) for cloud ice (column 1), snow (column 2), and graupel (column 3) particles, averaged over the total simulation time. Dust case minus Clean case differences for convective and stratiform data are presented in row 2 and row 4, respectively. Particle fall rate is determined by combining calculated particle terminal velocities (positive downwards, Khain and Sednev, 1995) with vertical velocity (positive upwards). In Fig. 8 (row 1 and 3), positive numbers indicate motion towards the surface, while negative numbers indicate the opposite."

>>Addresses changes to cloud ice fall speed and terminal velocity for the different cases at different altitude ranges. Cloud ice is generally heavier (greater terminal velocity) in the dust cases but is more strongly affected by vertical motion.

"Due to their small sizes, cloud ice fall rate is most affected by changes in vertical motion. The negative values of cloud ice fall rate in the convective average (Fig. 8 row 1a) at altitudes between 3 and 13 km indicate that transport of these small particles is predominantly upwards within the convective updrafts. The associated difference plot (Fig. 8 row 2a) indicate that fall rates between 3 and 9 km are increased for the primary dust cases (D.12; D1.2a; D12), and occur in conjunction with the greater updraft speeds at these altitudes. Cloud ice terminal velocities increase by approximately 0.4 (D.12), 1.7 (D1.2a) and 2.6 (D12) cm s$^{-1}$ between 3-9 km, which signify that the ice particles are becoming heavier due to increased diffusional growth.  Above 9 km the fall rates are increased in the D1.2a and D12 cases due to stronger downdraft intensities, as cloud ice terminal velocities are actually reduced by as much as ~1 cm s$^{-1}$ in the dust cases. In the D.12 case cloud ice fall rate and terminal velocities are reduced above 9km when compared with the clean case, while homogeneous freezing is reduced (Table 3). It is reduced by less than one percent in the D.12 case, indicating that many drops are still being transported to temperatures below -38°C and being frozen. While small drop sizes are not necessarily more common in the D.12 case (Fig. 6f), they are also not as significantly reduced as in the D1.2a (Fig. 6g) and D12 (Fig. 6h) cases, leading to a greater number of small ice forming homogeneously near the cloud top and reducing the average cloud ice fall rate. A similar reduction in cloud ice fall rate above 9km is seen in D1.2c. The added CCN in the D1.2c case increases the midlevel liquid content and results in a slightly higher homogeneous freezing number than the base D1.2a case (Table 3). Terminal velocity of the resulting homogeneous cloud ice particles is also slightly reduced compared to the D1.2a case due to

the generally smaller sizes of the frozen liquid drops. At altitudes between 3 and 9 km, terminal velocities are nearly identical in both D1.2a and D1.2c cases, with the D1.2c being slightly higher due to increased particle growth. The noticeable difference in fall rate between 6 and 9 km for the D1.2a and D1.2c cases is a result of stronger updrafts in the D1.2c case due to the greater latent heat release from the higher condensate mass. When deposition freezing is removed from the simulation (D1.2b), midlevel ice formation is provided by the immersion freezing mechanism. As this mechanism freezes liquid drops from the largest sizes to the smallest sizes, the larger drops freeze into graupel rather than cloud ice. Terminal velocities of cloud ice in the D1.2b case is reduced 1-3 cm s$^{-1}$ between 3 and 9 km, which indicates that the large change in fall rate is due to changes in the latent heat profiles affecting vertical motion."

>>Specifies changes to fall rates for snow and graupel. Snow and graupel fall rates are generally reduced between 5 and 10 km and increased above 10km for the primary dust cases. This helps to explain the lowered cloud tops and greater midlevel cloudiness.

"Changes in fall rates of precipitation sizes particles such as snow and graupel will strongly affect eventual surface precipitation accumulation due to changes in downdraft and below-cloud particle residence times and subsequent evaporation. Snow particles tend to grow larger by aggregation processes at warmer temperatures due to greater "stickiness" (Hallgren and Hosler, 1960), which results in fall rates generally increasing towards the surface (Fig. 8 row1b and row 3b). The greater midlevel ice formation in the dust cases results in increased aggregation rates in the 0°C to -38°C temperature range. Larger sizes settle out more quickly and tend to accumulate around the melting level as can be seen between 2 and 5 km. Terminal velocities are increased +15 (D.12) to +60 (D12) cm s$^{-1}$ over this range and are partially countered by stronger updrafts as can be seen in the fall rate differences (Fig. 8 row 2b). At altitudes between 5 and 10 km, snow terminal velocities are decreased between -5 (D.12) and -15 (D12) cm s$^{-1}$ due to more numerous but smaller aggregates, while the resulting fall rates vary between +2 (D.12) to -2 (D12) cm s$^{-1}$ due to the stronger vertical motion in these cases. Above 10 km, terminal velocity and fall rates are increased for the primary dust cases, although D.12 is reduced. In the D.12 case. the most significant ice formation and subsequent aggregation occurs primarily at homogeneous temperatures, yielding smaller aggregates near the cloud top. In the stratiform regime, where vertical motion is weaker, changes in terminal velocity are similar to the convective regime, but are higher overall. This is a result of more active aggregation in the stratiform regime due to the relative lack of liquid water content (for riming) compared to the convective core. When liquid content is significant, graupel forms either by direct freezing of large drops or by riming of existing ice and snow particles. In the primary dust cases (D.12, D1.2a, and D12), stronger updrafts and smaller graupel particles result from the greater midlevel ice concentrations. The graupel fall rates are progressively reduced between 3 and 10 km in both the convective and stratiform regimes as IN concentration is increased. In the D1.2b case, where immersion freezing results in significant formation of graupel from large frozen drops, graupel fall rates are significantly increased. This results in a final accumulated surface precipitation value in the D1.2b case which is 3.7% higher than the Clean case. In contrast, the final values of surface precipitation accumulation are reduced in the primary dust cases (D.12; D1.2a; D12), with the greatest reduction being -6.02% in the D12 case."

26) P15, L25-26: While Saharan dust may undergo less atmospheric processing than Asian dust, it still can serve effectively as CCN, as shown by Twohy et al (2009), the paper referenced by the authors in the previous paragraph. It might be more beneficial to write this paragraph in terms of representing the

thresholds of dust activation as IN and then at CCN and IN, where the studies presented up to this point represent the lowest end of CCN activation.

**We have restructured the paragraph P15,L18-29 to acknowledge that a study of pure IN effects represents the upper boundary of sensitivity to heterogeneous ice formation for a given dust concentration. We also reiterate the potential effects of dust acting as CCN.**

27) P15, L26-28: The authors mention that prior figures show that the simulations reproduce cloud effects witnessed in a suite of observational studies. Can the authors specify which cloud effects they are referring to (i.e., lower cloud top heights in dusty conditions)? Also, this statement does raise an interesting question regarding the utility of the model. If it is known that dust serve as both CCN and IN, and yet the model output compares well with the observations when dust is only allowed to serve only as IN, how well is the new parameterization actually performing? One way in which to address this would be using measurements of the CCN and/or IN activation capabilities of dust during this field campaign. Either way, a comment should be made in this regard in the manuscript.

**There are no measurements about the CCN and/or IN activation capabilities of dust during this field campaign.**

**We have expanded statement on P15,L26-28 from "As shown in previous figures, this IN only activation setting does allow for the model to reproduce similar large scale effects on the cloud field as those observed in Min et al. (2009) and Min and Li (2010)."**

**To : "As shown in previous figures, this IN only activation setting does allow for the model to reproduce similar large scale effects on the cloud field as those observed in Min et al. (2009) and Min and Li (2010), such as lowered cloud top heights, increased midlevel IWC, and similar changes to ice particle radii at different altitudes."**

28) P16, L19: Should "ice to liquid" be "liquid to ice"?

**The statement should have read "liquid to ice".**

**The order has been corrected in the text**

29) P16, L17-19: "The smaller liquid number at temperatures above freezing can be explained by the reduced ice content immediately above the melting level." Can the authors provide an additional sentence explaining the microphysical process that is causing the smaller liquid number here?

**We have added the statements to the end of P16,L17-19: ". . . immediately above the melting level. In addition to fewer small crystals forming near the freezing level and subsequently melting, the removal of small ice formation in the D1.2b case delays cloud glaciation to higher altitudes where immersion freezing becomes more prevalent. The reduced midlevel ice content results in a deeper liquid layer which enhances drop growth by autoconversion processes. The larger drops that form in this case efficiently remove liquid drops from the atmosphere by collision-coalescence processes or, when the drops are frozen into large graupel, small droplets will be removed by riming. Together, the reduced small ice melting and increased droplet collection result in the decreased total number of liquid drops at temperatures above freezing."**

**We have also clarified the P16,L19-20 sentence from: "The liquid PSD (Fig. 8e) is enhanced by the reduced conversion of ice to liquid, with more competition between the collecting drops shifting the size range slightly smaller." to "The liquid PSD (Figure 9e) at sizes larger than 81 um is enhanced by the reduced conversion of liquid to ice, with more competition between the collecting drops shifting the peak size range slightly smaller."**

30) P16, L33: From Figure 8c, the D12 case looks to be most different from the other dust cases, not D1.2b as stated in the text. If the authors are only comparing the D1.2a-c cases with this statement, then the authors need to specify this?

**We have changed the statement: "The D1.2b case exhibits the most distinct changes from the other Dust cases . . . " to:**

**"While the D12 case exhibits the greatest total changes in cloudiness over the vertical range, these changes are largely proportional with changes in IN number concentration. Specifically, the D.12, D1.2a, and D12 cases possess similar changes in percentage plot line shape that increase in magnitude as IN number increases. The D1.2b case exhibits the most distinct changes from the other Dust cases in terms of the shape of the resulting percentage line curves, due to the very different vertical profile of ice formation that occurred when deposition-condensation freezing was deactivated."**

31) P17, L3-4: The authors use domain-averaged OLR, but can the authors show OLR only from convective/stratiform columns? Given that cloud occurrence is only ~5% of domain (Figure 8c), it is difficult to justify that the domain-averaged values are represented of the changes to the cloud system.

**We have replaced the plot of domain averaged OLR with an average over the convective/stratiform cloud columns as suggested and have updated the text to reflect this.**

Figures and Tables

Most of the figures require improvement. The fonts are typically too small, and the sizes of each panel need to be increased. Line thicknesses also should be increased.

**Figures have been reworked to improve readability**

1) Figure 1: The color schemes don't seem to match in the two images, despite the fact that the images use the same colorbar. Can you please describe why there is this discrepancy?

**The observation and model data were initially rendered separately, due to a bug in the package related to the input datasets. An updated version of the visualization package has since been used to replot both datasets, which has corrected the color disparity in the figure.**

2) Figure 3: What are the lower lines in figure 3f, g and h? Are these the strongest downdrafts? This should be made clear in the caption. Also, are these counts of updraft grid points for the whole time period shown in the other figures?

**The lower lines in 3g-3h are the indeed the maximum downdraft velocities. The updraft grid point number covers the total integration time**

**We have clarified this in both the text and caption.**

3) Captions are needed for all of the tables in the text.

**We have added captions for each table.**

4) Figure 4: subfigure letters do not match subfigure letters described in text.

**Figure 4 subfigure letters corrected in text.**

5) Figure 5: the titles of the second and third rows do not match the caption. Please correct.

**Figure 5 caption corrected to match figure.**

6) Figures 7 and 9: The lines need to be made more distinguishable and thicker. This is especially the case for Figure 9a.

**Lines have been thickened in these and prior figures.**